# Higher socioeconomic status does not predict decreased prosocial behavior in a field experiment

James Andreoni [1], Nikos Nikiforakis [2 ✉] & Jan Stoop [3]

Does higher socioeconomic status predict decreased prosocial behavior? Methodological issues such as the reliance of survey studies on self-reported measures of prosociality, the insufficient control of relative incentives in experiments, and the use of non-random samples, have prevented researchers from ruling out that there is a negative association between socioeconomic status (SES) and prosociality. Here, we present results from a field experiment on the willingness of unaware individuals of different SES to undertake an effortful prosocial task—returning a misdelivered letter. Specifically, using the rental or sale value of homes as indicators of SES, we randomly selected households of high and low SES and misdelivered envelopes to them. Despite controlling for numerous covariates and performing a series of ancillary tests, we fail to find any evidence that higher SES predicts decreased prosocial behavior. Instead, we find that misdelivered letters are substantially more likely to be returned from high rather than low SES households.

[1] Department of Economics, University of California, San Diego, La Jolla, CA, USA. [2] Division of Social Science, and Center for Behavioral Institutional Design, New York University, Abu Dhabi, United Arab Emirates. [3] Erasmus School of Economics, Erasmus University Rotterdam, Rotterdam, The Netherlands. ✉email: nikos.nikiforakis@nyu.edu

The growing concentration of resources among the wealthy[1,2] and their mounting influence on public policy[3] has ignited a discussion about whether individuals of higher socioeconomic status (SES) are less prosocial than others. Early findings from psychological research received considerable media attention for suggesting there exists a negative relationship between SES and prosocial behavior[4–6]. SES refers to the social standing of an individual or a group in terms of income, education, and occupational prestige[5]. To explain the link with prosociality, psychologists proposed that affluence may be linked with reduced empathy and poverty with heightened empathy[4,7]. The balance of evidence, however, appears to have shifted since then. Scholars questioned the external validity of the early results on methodological grounds[8–10], and two attempts[11,12] to replicate the findings in ref. [5] failed. More recently, studies have either found no clear link between SES and prosociality[9,13,14] or, in the majority of cases, a positive association[15–20]. A positive relationship between income and prosocial behavior is also found in studies using large, nationally representative samples from around the world[10,21,22].

While, at first pass, the current balance of evidence would seem to suggest there may exist a positive relationship between SES and prosociality, upon reflection, one may have reservations about the correct interpretation of the data. The reliance of survey studies on self-reported measures of prosocial behavior may lead one to overestimate the relative prosociality of high SES people in a society if the latter feel under greater pressure to appear more prosocial than others. The insufficient control of relative incentives in experiments could have a similar effect as high SES individuals may give away more money not because they are more prosocial, but because they need the money less than low SES people who may be poorer. In other words, the measures used in previous studies cannot rule out alternative explanations for the findings. A methodological approach that would overcome the aforementioned concerns could be of great value as beliefs about the relative lack of prosociality of high SES individuals have been linked to preferences for taxing the wealthy at a greater rate[23], and heightened social tensions[24] which can undermine the smooth operation of institutions[25], economic growth[26], and financial development[27].

Here, we present evidence from a field experiment using the misdirected letter technique[28] to explore the link between SES and prosociality. Specifically, we used the rental and sale value of properties as a proxy for the SES of the individuals that inhabit them (see Supplementary Note 1). We then randomly selected households among those classified as being either high or low SES depending on their value, and (mis)delivered envelopes to them, i.e., we intentionally delivered envelopes to houses with the wrong address. To explore the link between prosociality and SES, we compare the rate at which high and low SES households forward the letters to the address written on the envelope. Returning an envelope is a costly prosocial act that is against one's (narrow) self-interest as doing so benefits another individual (the intended recipient), but requires time and effort[29,30]. These time and effort costs are likely to be small, implying that any differences across high and low SES groups are likely minimal. In line with this, survey respondents estimated they would need three minutes to repost a misdelivered letter (see Supplementary Note 5).

The experiment was conducted in a medium-sized city in the Netherlands—a country in which the existing evidence suggests there is no negative association between SES and prosociality[9,10]. For the main experiment, we randomly selected 180 high and 180 low SES households. High SES households in our sample have an average wealth of € 2,496,629. This is more than 90 times that of low SES households in our sample, € 27,237. For comparison, the average wealth in the Netherlands at the time of the experiment

was € 157,000. Given the central role of wealth in measuring SES and the size of the difference between groups, we believe we successfully identified households of high and low SES. Households were randomly assigned into treatments (see Supplementary Note 2). Specifically, to explore return motives, we use semi-transparent envelopes commonly used by the mail company, and vary their contents across different conditions. All the envelopes included a handwritten postcard from a grandfather to his grandson. In addition, depending on the treatment, the envelope contained either a banknote (that is, cash) or a bank-transfer card (BTC) of equivalent value that could only be claimed by the intended recipient (see Supplementary Fig. 2). In addition, we collected data for three ancillary treatments from different households, and two surveys to help interpret our findings.

Our methodology has several advantages. First, since individuals are unaware their choices are monitored, our method avoids demand effects and social desirability bias that could affect people of high and low SES differently[31,32]. Second, not only do we observe actual prosocial behavior, but also, in the case of envelopes containing BTCs, the prosocial act is comparable and equally attainable for everyone, irrespective of SES. Third, our research design avoids self-selection into the experiment which can bias estimates[32], even if samples are representative of observable characteristics as in refs. [10,21,22]. Self-selection can also result in certain treatment cells having small samples (e.g., for high SES), thus reducing statistical power. Fourth, we are able to obtain household-level data from CBS Netherlands for our sample on a number of key socioeconomic variables (e.g., wealth, income, household size). This allows us to check how wealthy a household is, perform randomization checks, explore the underlying differences between households of different SES, and investigate the robustness of our findings by adding controls in our regressions. Of course, our methodology is not without its limitations such as the fact that, due to the experiment being resource-intensive, we only consider one measure of prosociality in one city. We address this issue in the last section of our paper.

If affluence is negatively associated with the ability to empathize with others[4,7] and empathy is an important determinant of one's willingness to behave prosocially[33], then one would expect to find a negative relationship between SES and prosociality, all else equal. But all are likely not to be equal: high SES also implies better education, better employment, higher social status, etc. Each of these factors can affect prosocial behavior independently. In addition, these factors can interact with each other making it exceedingly difficult to identify mechanisms and offer a comprehensive account for the relationship between SES and prosociality. As noted in ref. [9], this relationship is "far from a simple pattern; it is a complex mosaic." The main aim of our experiment is to test the existence of a relationship between SES and prosociality, using a method that overcomes some of the interpretation issues found in previous studies. While a comprehensive account for any differences in prosocial behavior between high and low SES individuals is beyond the scope of our paper, the data may potentially permit us to explore two hypotheses about why SES may be positively associated with prosociality. The first is that, when the prosocial act involves monetary transfers, people with high SES behave more prosocially because they need the extra money less than the less well-off. The second is that because individuals with little wealth experience greater financial pressure[34] it may be more difficult for them to engage in acts that benefit others.

## Results

**Do high SES households return fewer misdelivered letters?** As mentioned, the rental and sale value of properties was used to

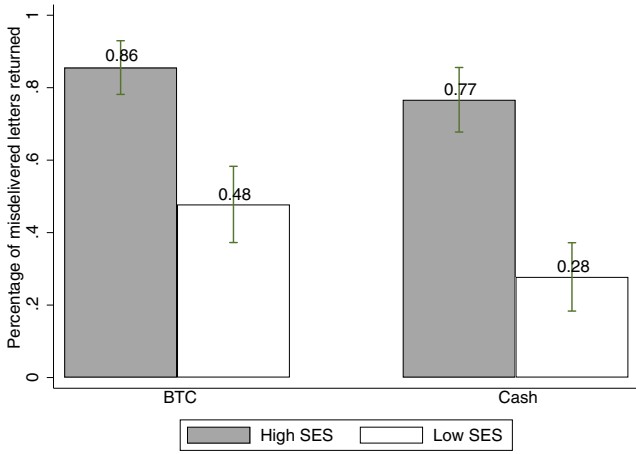

**Fig. 1 The figure shows the percentage of envelopes returned across conditions, error bars present 95-percent confidence intervals ($N = 90$ for each bar).** BTC refers to the treatment in which the envelope contains a bank transfer card of the same value. Cash refers to the treatment in which the envelope contains a banknote. Across treatments, high SES individuals return significantly more envelopes ($N = 360$, $p = 0.000$, two-tailed, Fisher-exact). The same applies for the BTC treatment ($N = 180$, $p = 0.000$, two-tailed, Fisher-exact), as well as for the Cash treatment ($N = 180$, $p = 0.000$, two-tailed, Fisher-exact).

categorize houses as high/low SES (see Supplementary Note 1). Figure 1 shows the percentage of envelopes returned in each treatment by SES. Across treatments, high SES households are more than twice as likely to return misdelivered envelopes than low SES households (81% vs. 38%; $N = 360$, $p < 0.01$, two-tailed, Fisher-exact; see Supplementary Data 1, and Supplementary Code 1). In the Cash treatment, 76.7% of the envelopes misdelivered in high SES households were returned, compared to 27.8% of the envelopes misdelivered in low SES households ($N = 180$, $p < 0.01$, two-tailed, Fisher-exact). This difference could be due to the fact that high SES households are wealthier and, hence, need the money-less. A similar pattern, however, is observed in the BTC treatment where 85.6% of the envelopes misdelivered in high SES households are returned, and 47.8% of the envelopes in low SES households ($N = 180$, $p < 0.01$, two-tailed, Fisher-exact). Figure 1 also reveals that low SES households are substantially less likely to return envelopes containing cash than BTCs (47.8% vs. 27.8%; $N = 180$, $p < 0.01$, two-tailed, Fisher-exact) relative to high SES individuals (85.6% vs. 76.7%; $N = 180$, $p = 0.18$, two-tailed, Fisher-exact). We reach the same conclusions if we disaggregate the data to condition the analysis on the BTC amount or the amount of money inside the envelope (see Supplementary Note 3).

**Exploring the determinants of prosociality.** To explore in greater depth the determinants of returning envelopes, we turn to regression analysis. The empirical model estimates coefficients for our main variables of interest (High SES, Cash) while controlling for a number of covariates that could affect return rates. The first of these controls is the number of weeks since the last payday at the time of the misdelivery, and its interaction with High SES (Week, Week × High SES). Specifically, in the Netherlands, wages and unemployment benefits are typically paid in the last week of each month. Since financial pressure can affect the behavior of low SES individuals, especially as money runs low[34], we hypothesize that envelopes misdelivered in low SES households (but not high SES households) will be less likely to be returned as time passes from the last payday (hence the interaction term). The analysis also controls for the distance of the house from the

nearest mailbox (Distance Mailbox) and from the house of the intended recipient (Distance Recipient's House). We hypothesize that the larger the distance between the participant's house and the public mailbox or the house of the intended recipient, the greater the physical effort required to return a letter and, hence, the lower the probability a letter is returned. We also include a control capturing income inequality at the neighborhood level. This information was obtained by CBS Netherlands, which, however, did not have this information for 11.6% of our sample (see Supplementary Note 4). While in light of the findings in refs. [22,35], and ref. [36], we do not expect that this variable moderates our findings, we control for income inequality to check for robustness. Finally, to explore the possibility of a social-comparison effect[37], specifically, that relative wealth is associated with prosocial behavior[38], our analysis controls for relative income at the district level[39]. Standard errors are clustered at the street level to account for possible neighborhood effects.

The results from the regression analysis are presented in Table 1. Models (I)–(III) present our main findings. Model (I) shows that, across treatments, the likelihood that high SES households return an envelope is 113% (=$100 \times 0.43/0.38$) of that of low SES households. As can be seen in models (II) and (III), the low SES group is 20 percentage points less likely to return envelopes containing cash than BTCs. The interaction term Cash × High SES shows that the high SES group is 11 percentage points more likely to return cash envelopes than the low SES group (relative to BTC envelopes). Although this difference is not statistically significant, the relative reduction in return rates between BTC and Cash is far greater for low SES households (71.9%) than it is for high SES households (10.3%). This hints toward the greater need that low SES households have for money. The coefficient for Week in the model (III) reveals that financial pressure seems to negatively affect the probability that low SES households return envelopes. Consistent with this interpretation and our hypothesis, no such trend is detected for high SES households ($p = 0.92$). Model (III) also shows that, as hypothesized, the longer the distance between a participant's house and the public mailbox, the lower is the probability that the letter is returned to its intended recipient.

The findings in Models (I)–(III) in Table 1 show that, once we add controls, the relative return rates for high and low SES households drop from 113% in Model (I) to 79% in Model (II) and 48% in Model (III). This illustrates the importance of controlling for the relative incentives across SES, as well as the financial pressure low SES individuals, maybe under when making inferences about their motives. Despite this, the high SES group is still found to be 31 percentage points more likely to return BTC envelopes than the low SES group in the model (III). A number of factors could account for this sizable difference such as differences in wealth, education, and social status between high and low SES households. While our data do not permit us to identify the mechanism behind the difference, the evidence is clearly at odds with the early findings in psychological research that suggested the existence of a negative relationship between SES and prosociality.

**Robustness checks.** Models (IV)–(VII) in Table 1 provide robustness checks for our findings. Specifically, model (IV) re-estimates model (III) after dropping from our sample households that include non-Dutch members to rule out the possibility that differences in literacy or in empathy toward foreigners affect return rates. As can be seen, apart from the constant term, the estimates are largely unchanged in this sample. Model (V) again re-estimates model (III) using only observations from single-person households to rule out the possibility that the greater

**Table 1 The determinants of return envelopes.**

| | (I) | (II) | (III) | (IV) | (V) | (VI) | (VII) |
|---|---|---|---|---|---|---|---|
| High SES | 0.43*** (0.04) | 0.38*** (0.07) | 0.31*** (0.07) | 0.28*** (0.07) | 0.22* (0.13) | 0.28*** (0.09) | 0.28*** (0.08) |
| Cash | | −0.20** (0.09) | −0.20** (0.09) | −0.21** (0.08) | −0.24** (0.1) | −0.20** (0.09) | −0.19** (0.09) |
| Cash × High SES | | 0.11 (0.09) | 0.11 (0.1) | 0.13 (0.09) | 0.01 (0.23) | 0.13 (0.1) | 0.12 (0.1) |
| Week | | | −0.08* (0.03) | −0.09** (0.03) | −0.09** (0.03) | −0.08** (0.03) | −0.07* (0.03) |
| Week × High SES | | | 0.08* (0.04) | 0.09* (0.04) | 0.08 (0.1) | 0.08* (0.04) | 0.08* (0.04) |
| Distance recipient's house | | | −0.01 (0.01) | −0.02* (0.01) | −0.01 (0.05) | −0.02 (0.01) | −0.01 (0.01) |
| Distance mailbox | | | −0.14** (0.05) | −0.15** (0.05) | 0.06 (0.17) | −0.28* (0.16) | −0.12 (0.08) |
| Gini | | | | | | 0.20 (0.63) | |
| Relative income | | | | | | | 0.07** (0.03) |
| Constant | 0.38*** (0.03) | 0.48*** (0.06) | 0.64*** (0.08) | 0.69*** (0.07) | 0.59*** (0.20) | 0.64*** (0.19) | 0.54*** (0.09) |
| N | 360 | 360 | 360 | 333 | 143 | 318 | 341 |
| Sample restrictions | None | None | None | Dutch HH | 1 Person HH | None | None |
| R2 | 0.19 | 0.22 | 0.24 | 0.24 | 0.13 | 0.24 | 0.26 |

Estimates are from a linear probability model. The dependent variable takes the value of 1 when an envelope is returned and 0 otherwise. High SES and Cash are dummy variables indicating whether an observation is associated with a high SES household or the Cash treatment, respectively. Week measures the number of weeks since the last payday. Distance Recipient's House and Distance mailbox are the driving distance in km from the household where the envelope was misdelivered to the intended recipient's home and to the nearest mailbox, respectively. Gini is the Gini coefficient of the neighborhood of the subject's household. Relative Income is calculated as the household income per member, divided by the average household income per member in the district. Standard errors are shown in parentheses (clustered at the street level). The results are virtually identical when we use a Probit specification. Reported results are from two-tailed tests.
***, **, * indicate significance at the 0.01, 0.05, and 0.10 levels, respectively.

propensity of high SES households to return envelopes is somehow associated with differences in household sizes (e.g., audience effects). This also ensures that the person returning the envelope is the income-earner and not someone else living in the household. Although some of the estimates differ, our findings hold qualitatively, despite using less than 40% of our sample. Model (VI) extends model (III) by adding a control for income inequality at the neighborhood level. As can be seen, adding this control leaves our estimates largely unchanged. Finally, model (VII) shows that our main findings are largely unaffected when controlling for relative income. The latter has a positive effect on return rates. Given the private nature of the prosocial act in our experiment, one possible interpretation for this finding is that higher relative income is associated with feelings of social responsibility. Additional robustness checks can be found in Supplementary Table 3 where we use more of the information that we obtained from CBS Netherlands.

**Ancillary tests**
The difference in return rates for high and low SES households need not reflect exclusively differences in the prosociality of their members. For this reason, we performed a number of ancillary tests to see if we can find any evidence suggesting that SES is negatively associated with prosociality. As we will see, we failed to find such evidence.

The simplest way to return misdelivered letters is to drop them into one of the public mailboxes on the street. Patterns consistent with our data could emerge if mailmen working in low SES neighborhoods were substantially more likely to retain our envelopes. To check this, we collected data for a control condition in which we posted letters for all experimental treatments directly to the address of the recipient. This was done on the same days on which the other treatments were administered, and from the same neighborhoods as the houses in our sample. Only one out of 85 envelopes were not delivered as posted. We can therefore rule out this possibility.

Similar differences between high and low SES households could also be observed in individuals in the low SES group who check their mailboxes less frequently or are less knowledgeable about how to efficiently return misdelivered letters. To explore these possibilities, we conducted a survey with individuals renting housing from the city's social-housing corporation ($N = 45$; see Supplementary Data 2, and Supplementary Note 5). On average, respondents report checking their mailboxes six times per week, and 75.5% of them report checking it daily. Further, 89% of them were aware they could return misdelivered letters simply by using the public mailboxes on the street. If, for robustness, we assume that only 89% of low SES households in our sample know how to return a misdelivered envelope, whereas 100% of high SES households do, the difference in return rates between them remains highly statistically significant in all treatments ($N = 340$ or 170, $p < 0.01$, two-tailed, Fisher-exact). This explanation, therefore, can also not account for our results.

It is also possible that individuals in low (high) SES households recognized the intended recipient's address as being in a middle-class neighborhood and thus perceived him as being better-off (worse-off) than they are, making them less (more) likely to return the envelope. To test this explanation, we collected data for an additional treatment, in which we misdelivered envelopes with BTCs to low SES households, but altered the message on the postcard to reduce the perceived social distance: "Dear Joost, I understand you are still waiting for a social housing apartment to become available and you are having trouble paying your bills. Here is € 20 for you. I know it's not much, but it is all I can afford —Your grandfather." The return rate is 53%, which is similar to

the 47% in the BTC treatments ($N = 135$, $p = 0.59$, two-tailed, Fisher-exact), suggesting that this effect is not strong enough to account for our findings.

Another explanation is that trust in the ability of the mail company to deliver the envelopes is negatively associated with SES. Indeed, Dutch respondents to the World Values Survey[40] report having less trust in institutions such as the government, the police, and the press as wealth decreases ($p < 0.01$, two-tailed, Ordered Probit, for all tests). Could this extend to the postal service? We conducted a survey with Dutch-only students at Erasmus University ($N = 133$). Our sample includes respondents from a broad range of socioeconomic backgrounds, from the lowest to the highest SES families (see Supplementary Data 3, and Supplementary Note 6). After adjusting our estimates using population weights, we find a positive correlation between SES and trust in the government ($p < 0.10$, two-tailed, Ordered Probit), the police ($p < 0.05$, two-tailed, Ordered Probit), and the press ($p < 0.05$, two-tailed, Ordered Probit). However, we do not find a relationship between SES and trust in the postal service ($p = 0.54$, two-tailed, Ordered Probit). Therefore, this hypothesis also seems unable to account for the difference in return rates seen in our main experiment.

Finally, we consider the possibility that reasons other than trust may prevent low SES households from returning the letters. To that end, we collected data for a treatment in which we provided monetary incentives to post a letter. Specifically, we delivered letters to high and low SES households promising to pay € 20, if an enclosed envelope was returned within a month (see Supplementary Note 7). We justified this request by saying that we were studying the use of postal services. Low and high SES households posted 60% and 69% of the envelopes, respectively ($N = 90$, $p = 0.51$, two-tailed, Fisher-exact). The difference in return rates in this and the BTC treatment is $-16.67\%$ for high SES and $+12.22\%$ for low SES. This means that whereas low SES households return more envelopes when it benefits themselves, high SES households return fewer. The 28.89 percentage point difference-in-difference is substantial and statistically significant ($N = 90$, $p < 0.01$, two-tailed, linear probability model). This hypothesis, therefore, also seems unable to account for the difference in return rates in our main experiment.

## Discussion

We have presented results from a field experiment testing the existence of a relationship between SES and prosociality. In contrast with early reports of a negative association between SES and prosociality, we find that high SES households behave more prosocially than low SES households across all conditions in our study. A series of ancillary tests failed to provide any evidence that could suggest the existence of a negative relationship between SES and prosociality.

Although no empirical study can provide a definitive answer to a question as general as to whether individuals of high SES tend to behave more (or less) prosocially than low SES individuals by itself, we believe our findings make an important contribution to advancing our understanding of the topic. Specifically, our methodology overcomes the identification issues present in previous empirical studies. By showing that high SES households are not more likely to behave less prosocially than low SES households in a natural setting—thus avoiding demand effects and social desirability bias—while controlling the relative incentives for behaving prosocially (see BTC treatment) and avoiding issues of self-selection, our study helps alleviate concerns about how the fast-growing body of evidence from surveys and incentivized experiments contradicting the existence of a negative relationship between SES and prosociality should be interpreted. In summary,

while more field experiments are clearly needed, in light of our findings, the current balance of evidence would seem to suggest that SES does not predict decreased prosocial behavior as early findings in psychological research suggested.

The evidence that SES is not negatively associated with prosociality also contradicts a stereotype that is common in many countries across the globe[41]. According to it, the wealthy are excessively selfish[42–45]. While stereotypes, i.e., widely held but oversimplified ideas of a particular type of person[46], can economize on information-processing costs[46–48], they can be detrimental for welfare if they are inaccurate. Indeed, the stereotypical view of high SES individuals has been linked to preferences for taxing the wealthy[23] and heightened social tensions[24].

The methodology used in the present study, of course, is not without limitations. A field experiment such as ours is resource-intensive. As a result, it cannot easily be scaled to nationally representative samples or to consider different kinds of prosocial behaviors. This raises a question that is all-too-familiar to experimental social scientists: To what extent can we expect the findings from this specific experiment to generalize (i) to different social contexts, and (ii) to different populations? With regards to (i), the propensity to return envelopes in misdirected-letter experiments has been linked with giving in dictator games[29]. The latter is arguably the most commonly used method in incentivized experiments for measuring altruism, i.e., the extent to which an individual cares about the welfare of others[4,5,10,13,15–17,21,22,49]. Altruism, in turn, has been linked to the propensity of individuals to volunteer, help strangers, and donate money[21]. With regards to (ii), researchers have previously used surveys and incentivized experiments to study the relative prosociality of wealthy people in the Netherlands, where our experiment also took place. They did this using nationally representative samples[9,10], and different measures of prosociality[9,10,17]. Like with our experiment, none of these studies found evidence that high SES individuals behave less prosocially than low SES individuals. In fact, on balance, the evidence shows that high SES individuals behave more prosocially: they are not more likely to lie, steal, or accept bribes[9], but they volunteer more, trust more, help more, and give more to charity[9,10] than those with lower SES. Similar findings are obtained in representative samples in other countries around the world[10,21,22]. Taken together, therefore, these findings suggest that there are no obvious reasons to expect that our results will not generalize to other social contexts or populations. Of course, the question of generalizability is ultimately an empirical question that can only be answered with more field experiments.

Two previous field studies addressed questions related to ours. In ref. [4], individuals driving more expensive cars in California were found to be more likely to violate the traffic law. Apart from the fact that the act is not comparable for high and low SES individuals as the former are likely to find it easier to pay off traffic fines, the value of a car is an imprecise measure of wealth making the interpretation of the findings difficult. Another study hypothesizes and finds that letters "lost" (either dropped by the experimenters on the street or placed on the windshield of a car) in more affluent neighborhoods in the Netherlands were more likely to be returned[50] (see ref. [51] for a lost-letter experiment in Germany). Although the evidence in ref. [50] is in line with our findings, they do not permit inferences about the relative prosociality of the high and low SES individuals as it is unclear who finds the letter. This, in fact, was the original inspiration for the misdirected-letter technique[28] that we use in our experiment.

In conclusion, our study contributes to (and reinforces) a growing body of evidence suggesting socioeconomic class does not predict decreased prosocial behavior. The contrast between the stereotypical view of high SES individuals[41] and the current balance of evidence about their relative prosociality points toward

a great need for further research. This research program can play an important role in informing policies that may help reduce social tensions[24] and alter preferences for redistribution[23] with obvious socioeconomic implications. On the one hand, future studies can help understand the limits of our findings: under what conditions may high SES individuals behave less prosocially than others? Such studies will help uncover why wealthier individuals may sometimes behave more prosocially than others. For instance, there may be social returns to kindness in some societies, or there may be a causal link between wealth and norm compliance that could be mediated through more positive life experiences, better education, etc. On the other hand, future studies could attempt to shed light on the origins of the stereotypical view of high SES individuals. This will require the collection of detailed information on individuals such as the process through which they acquired their wealth[52,53], their level of education[9], their family background, and their beliefs. By shedding light on the origins of this stereotype, these studies will help us defeat our own biases and forge constructive solutions to the real challenges presented by the growing wealth inequality.

## Methods

**Experimental treatments and procedures.** All aspects of the study, including ethical acceptability, were reviewed by the Internal Review Board at Erasmus University Rotterdam. The review was conducted after the experimental data had been gathered as the Erasmus School of Economics did not have an IRB process at the time of data gathering. To determine whether participants had to be debriefed, Jan Stoop had a conversation with the Dutch mailing company after the experiment had been completed. The company decided that debriefing was not required.

In all treatments, a semi-transparent envelope was purposefully misdelivered to the mailbox of pre-identified households. All envelopes included an identical postcard with the following hand-written message visible on the back: "Dear Joost, here is €x for you—Your grandfather." Joost is a research associate of ours, living in a middle-class neighborhood. We chose a middle-class neighborhood to avoid ingroup/outgroup effects between high and low SES individuals, or other confounds such as differences in proximity. Joost's address was clearly visible on the front side of the envelope (obscured in Supplementary Fig. 2 for privacy reasons) which also contained a stamp and a postmark making it seem as if the envelope had been misdelivered by the mail company.

The experimental treatments vary the additional content of the envelopes. In the BTC treatment, we included a bank-transfer card (BTC) for the amount of either € 5 or € 20. A bank transfer card is an order to transfer money from one bank account to another. It is impossible to go to a bank office to cash it and, therefore, there is no monetary gain from keeping an envelope with a bank transfer card. In the Cash treatment, the envelope included a banknote of either € 5 or € 20. Across treatments, we find that the amount in the envelope (i.e., € 5 or € 20) has no economically or statistically significant effect on behavior. For brevity, we pool observations across amounts. The disaggregated analysis can be found in Supplementary Fig. 1.

A feature of our study is that we link our experimental data to socioeconomic data provided to us by CBS Netherlands. Before conducting our experiment, we contacted CBS who consented to us using their data. To ensure the privacy of participants was not compromised, it was agreed that, once we had collected the experimental data, we would provide CBS with our dataset. CBS would then replace all household-identifying information with a Random Identification Number (RIN) which would allow us to match our data to theirs, not needing household-identifying information such as street address. Importantly, the CBS data could not be exported outside of a special virtual environment created by CBS for the purpose of data analysis.

In order to track which households returned envelopes, we recorded the serial numbers of the banknotes or the numbers on all bank transfer cards prior to the (mis)delivery. Letters were always (mis)delivered on weekdays at the same time the mail company makes deliveries, between 1:00 p.m. and 2:00 p.m. To avoid suspicion, misdeliveries were always done in the uniform of the official mail company. To control for day-of-the-week and week-of-the-year effects, we misdelivered an equal number of envelopes for each treatment each time a round of misdeliveries was done. There were seventeen rounds in total. All observations were collected over a 10 week period, between early October and mid-December.

Finally, to determine the weeks since a household's last payday we relied on the fact that those in our sample on public assistance are paid either on the 22nd or the 23rd day of each month. While we do not know exactly when high SES individuals in our sample are paid their salaries, it is common in the Netherlands for salaries and pensions to be received around the same dates.

**Selecting households.** All households were located in the same medium-sized city in the Netherlands. Households were randomly selected through a two-step process. In the first step, to maximize the probability of selecting either high or low SES individuals to participate in our experiment, we compiled two lists of households following a procedure detailed below. In the second step, we randomly selected households from these lists for participation.

Our first concern was to compile a list of high and low-SES households. For this reason, we decided to target some of the wealthiest and poorest households in the city. For safety and privacy concerns, no institution was willing to share with us a list that would be linking individual addresses with household wealth, prior to data collection. CBS Netherlands was willing to provide us with data on household wealth after the experiment was conducted. This way, they could encrypt the data, guaranteeing the privacy and anonymity of all participants. For this reason, we had to compile our own list of households to select from using a measure that would be highly correlated with wealth. We used the property value of one's house to identify high SES individuals, and the rental price to identify low SES individuals.

To select low SES households, we took advantage of the fact that the city has several social-housing corporations. For one of these, the explicit purpose is to rent out apartments to the poorest people in the city. The social-housing corporation was kind enough to share with us a list with the addresses of their cheapest apartments, from which we selected 152 households. As this number did not suffice for our study, we selected additional apartments by using the website of the same corporation in which apartments are advertised for rent. In particular, we selected apartments in listed buildings, but not the apartments that were for rent.

After compiling the list of low SES households, we randomly selected 225 of them to participate in our experiment (180 used in our main experiment and 45 used in the "Low SES Joost" treatment), provided that two conditions were met. First, most apartment buildings have four stories and share entrances with eight apartments. Although an official from the social-housing corporation informed us that residents have little contact with each other, we decided to limit misdeliveries to three mailboxes per entrance to minimize communication between households. We randomly selected 159 households sharing an entrance with two other apartments in the same building, and 66 households that do not share an entrance. We find no economically or statistically significant differences across these two groups. Households sharing entrances returned 38.4% of misdelivered envelopes while households not sharing entrances in our sample returned 47.0% ($N = 225$, $p = 0.24$, two-tailed, Fisher-exact).

The second condition was that the family living in a selected household is native Dutch. The reason is that the ethnicity of the sender and receiver can influence prosocial behavior[54–57]. This could be a problem as non-natives in the Netherlands tend to be poorer. To avoid this confound, we took pictures of all family-name signs on doorbells and ensured that all apartments in our sample were likely to be native Dutch. Using data from CBS Netherlands, we can control for the effect of any remaining non-Dutch households.

To compile a list of high SES households, we started by consulting www.Funda.nl, a website in the Netherlands that advertises houses for sale. Our goal was to identify neighborhoods and streets where wealthy individuals live. To do this, we found all neighborhoods with a sale price of at least € 750,000. All houses on the same street as the identified houses were included on our list of high SES households, excluding the houses that were up for sale. Once this procedure was done, we consulted www.Postcode.nl. This website contains information on all houses in the city in which our experiment was conducted such as the postal code, house number, and surface. We used this website to include on our list all houses that were not for sale but had a comparable surface as those that were for sale.

Once the list was compiled, we randomly selected high SES households to deliver the misdelivered letter. To minimize the effects of communication between subjects, we excluded houses that were close to each other. A total of 100 houses in our sample have no neighboring house that is selected for any of the core treatments. The other 80 houses do have at least one neighbor that is a subject in one of the core treatments. (Almost all of these houses are villas with a large surface and driveways that are far removed from the driveway of the neighbor.) Differences in return rates between houses with a neighbor (76.3%), or without a neighbor (85.0%) are insignificant (Fisher exact test, $p = 0.18$).

**Power calculations.** The safeguards we agreed upon in order to limit the chance that those in our sample would meet each other and learn they are in an experiment allowed us to enroll up to 90 high SES and 90 low SES households per treatment (for a total of 360 households). This raises the question of whether we have sufficient statistical power to detect differences between high and low SES individuals. High SES people were found to be twice as likely as low SES individuals to choose the selfish over the prosocial action in ref. [5] [studies 1, 2, and 4]. If we assume that high SES households return 40 percent of the envelopes and that low SES households are twice as likely to return envelopes (i.e., 80 percent), to detect a difference at the 5-percent level of significance 80 percent of the time using non-parametric tests, we would need 23 high SES and 23 low SES households per treatment. Our sample is therefore 3.91 times as many as what we would need to find an effect similar in magnitude to that reported in[5].

**Reporting summary**. Further information on research design is available in the Nature Research Reporting Summary linked to this article.

## Data availability

The authors declare that the experimental data and the statistical code used in the analysis are available as supplementary information files. We are restricted in our ability to share the data obtained from CBS Netherlands. Interested parties can obtain this data directly from CBS Netherlands. The authors are willing to provide interested parties with information on how to obtain this data from CBS Netherlands.

## Code availability

The code used to generate the analysis is available from the Supplementary Information. It contains STATA 15.0 files with the raw data and the code used to perform the statistical analysis.

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

## Acknowledgements

The authors thank Olivier Bochet, Zachary Breig, Alain Cohn, Aurélie Dariel, Robert Dur, John Ham, PJ Henry, Hans van Kippersluis, Steven Levitt, John List, Drazen Prelec, Vitalie Spinu, Darjusch Tafreschi, Joost Verlaan, Pepijn Pest, and seminar participants at the University of Athens, the Berlin Social Science Center, the University of Chicago, Kiel University, the University of Nottingham, Tilburg University, and Wageningen University. We are grateful to ERIM (Vidi grant VI.Vidi.195.061), the National Science Foundation (grants: SES-1427355, SES-1658952), the Science of Philanthropy Initiative, the John Templeton Foundation, and Tamkeen under the NYU Abu Dhabi Research Institute (Award CG005) for financial support.

## Author contributions

J.A., N.N., and J.S. designed the experiment, prepared the surveys, carried out the statistical analyses, and wrote the manuscript, J.S. conducted the experiment, and compiled the dataset.

## Competing interests

The authors declare no competing interests.
