## [Peer Review File · Nature Communications]

Reviewers' Comments:

Reviewer #1:

Remarks to the Author:

Report

Rejecting the selfish rich stereotype in a field experiment

The article presents a field experiment (n=180 rich and 180 poor respondents in the main experiment) studying the question whether the common stereotype of selfishness of rich people is correct. The authors found clear evidence that rich people behave less selfish and hence much more prosocially than poor people.

While I like the design of the study, I have a couple of serious concerns and questions.

First of all, the method is innovative. However, the issue is obviously that a number of lab experiments and survey studies found inconsistent results. While this is really interesting, a field experiment does not solve these inconsistencies.

Rather, there is now one more study showing that the rich are not more selfish than the poor – under the given conditions. To arrive at a valid conclusion about what is going on, the existing studies should be much more neatly compared and evaluated. As far as I see, settings are considerably different as are the point of departures and the variables included. For example, the finding by, e.g. Coté, House and Willer (2015, mentioned also by the authors) is on giving in the presence of income inequality. This is quite a different angle. In the present study, no control for inequality is included (e.g. inequality of the neighborhood would have been an option).

So, the article fails to make clear why a field experiment can solve the issue - it actually does not. This is my main concern.

Minor points:

- the setting seems to be quite artificial: who sends cash in a semi-transparent envelope? I am surprised that this worked so well.
- The discussion about the costs of posting the letter is overstretched: there are many mailboxes. The behavior under study is a relatively small prosocial act towards an anonymous stranger.
- Why are the results for the 5 euro and 20 euro not presented separately? Also, effects of the control variables are just reported as insignificant but not shown. On the other hand, insignificant results (see the interaction effect in table 1) are reported and interpreted.
- The online supplementary material is already on the web and dates back to August 2017, and presents more extensive analyses. Also, I found the study (in a version of 2017) on EconWeb. Is the study already published? There is also a YouTube clip about the study, stressing actually that poor and rich differ only in their behavior but not in their preferences. See here
- The variables included in the regression model could be explained and justified more systematically. E.g., it is not argued in the theory section that the number of weeks since pay day is expected to matter.
- Is there also a letter sent without money, be it cash or bank transfer? Why are the results not reported?
- Are there foreigners who could read the letter? In poor areas the number of foreigners is higher.
- The ancillary study where respondents got monetary incentives for posting a letter does actually

show no difference between rich and poor. It also does not study postal services. It does study what happens if you benefit yourself (in contrast to others).

- Studies in the references 35) and 36) are wrongly cited, they are not about rich and poor neighborhoods. Also, these studies deliberately address passengers in a neighborhood and not households.

Reviewer #2:

Remarks to the Author:

There is some controversy in the literature regarding the relation between socioeconomic status and prosocial behavior. There is evidence that low SES persons are more prosocial than their high SES counterparts; as the current authors point out, there is also contrary evidence. There have been attempts to resolve this discrepancy in findings, arguing that the SES-prosocial behavior relation is moderated by factors such as degree of economic inequality, or whether the prosocial behavior in question is public or private in nature.

Rather than addressing these moderating factors, the current authors appear to cast doubt on the validity of prior findings (they mention small, convenience samples; self-reported behavior, etc., although such issues do not apply to all of the relevant prior research) and propose that their study is free of such issues. The study they report is certainly interesting and methodologically ingenious. Whether it settles the question of how SES is related to prosocial behavior is open to question, however.

Starting with the strengths of the research, the 'misdirected letter' methodology has the advantage of being unobtrusive, removing (or at least minimizing) the influence of social desirability factors. Furthermore, the use of translucent envelopes with readable messages and visible cash or bank transfer card (BTC) contents is a clever way to vary the personal and social incentives for re-posting the letter. The sampling of households by property value, verified by CBS household income data, is an effective way of ensuring that the households to which the misdirected letters were 'posted' differed markedly in wealth. The findings appear to be clear-cut: in all conditions, richer households were more likely than poorer households to re-post the misdirected letters, although there was some evidence that this difference was moderated by whether the envelope contained cash or a BTC.

Turning to the limitations of the study, my main concern is that this is a study conducted in one city in one country, using a very specific measure of prosocial behavior. The authors suggest (in the title of their paper) that their results reject the stereotype of the 'selfish rich'. In their Discussion, they go so far as to claim that "our findings indicate the stereotype of the selfish rich is inaccurate" – a very general claim that fails to acknowledge the specificities of their research.

To what extent does the misdirected letter technique measure prosocial behavior? The beneficiary is remote and the need for help is quite abstract. The measure may to some degree reflect differences in institutional trust (I am not convinced by the ancillary findings relating to this issue based on a sample of university students, who are unlikely to reflect the sociodemographic attributes of the study sample). More importantly, perhaps, given the claims in the paper, there is the issue of the extent to which 'selfishness' is captured by this measure. Is failing to re-post a letter a 'selfish' act, especially in the conditions where there was no possible benefit to self by not doing so (other than the saving in time and effort)? As Table 1 in the Supplementary Information shows, the rich households differed from the poor ones in several ways other than income/wealth: On average, the rich households are older, have larger families, are less ethnically diverse, and receive more pensions. I appreciate that the authors controlled for these (and other) differences in their regression analyses, but I doubt

whether such statistical controls can remove the effect of the rich households simply having more spare time available.

Then there is the issue of the specific location of the research: one medium-sized city in one European country, a country with relatively low economic inequality as measured by the Gini coefficient (28.6%, versus 47% in the USA, according to the World Bank). Given the evidence that the SES-prosocial behavior relation is moderated by inequality, this strikes me as a potentially serious limitation.

In summary, what we have here is a well-conducted study showing that in this specific social context, rich households are more likely than poor ones to re-post a misdirected letter. But before anyone can reasonably claim to be able to reject the stereotype of the selfish rich, evidence from a wider variety of contexts using a broader range of measures is surely needed.

Reviewer #3:

Remarks to the Author:

This is an excellent paper. It deals with a relevant and interesting question, is well written, straightforward, easy to comprehend, and supplemented with a number of ancillary tests. Congratulations to the authors! I enjoyed reading it. I have only a few minor comments and suggestions.

1) The results displayed in Table 1 are based on a linear probability model. Are the results comparable if the analyses are conducted via logit or probit models? Could you include a sentence in the caption of Table 1 indicating the similarity (or difference) of results?

2) Poor households often have a higher proportion of individuals with a migration background. The norm that misdelivered letters should be returned might be less widespread in such a population. Hence, you might compare two groups that would not only differ in wealth but also in their normative background. Can this possibility be excluded? If not (or yes) you might want to mention or discuss it shortly in the discussion section.

3) Also for the discussion section: The rich are obviously the winners in a given society. Therefore, it does not come as a surprise that they support the system, have higher trust in societal institutions, and show more norm conforming behavior, including following the norm to return a misdirected letter. Hence, higher norm compliance is a further candidate to answer the "why question".

Kind regards
Axel FRanzen

Response to the comments made by Reviewer 1

We are thankful for your careful reading of our manuscript and thoughtful comments. We have addressed them all. We hope you will be satisfied with the changes we made to the manuscript in response to your comments and that you will find the new version to be suitable for publication in *Nature Communications*.

Below, we present your comments in boxes, followed by our responses.

The article presents a field experiment (n=180 rich and 180 poor respondents in the main experiment) studying the question whether the common stereotype of selfishness or rich people is correct. The authors found clear evidence that rich people behave less selfish and hence much more prosocially than poor people.

While I like the design of the study, I have a couple of serious concerns and questions. First of all, the method is innovative. However, the issue is obviously that a number of lab experiments and survey studies found inconsistent results. While this is really interesting, a field experiment does not solve these inconsistencies. Rather, there is now one more study showing that the rich are not more selfish than the poor – under the given conditions. To arrive at a valid conclusion about what is going on, the existing studies should be much more neatly compared and evaluated.

(continued below)

We were delighted to learn that you liked the experimental design, and that you found our method to be “innovative” and the topic of the study “really interesting”. Your comment suggests that your main concern is whether our experiment advances our knowledge on the topic sufficiently to warrant publication at *Nature Communications*. In hindsight, we see how we could have explained better why our findings make an important contribution to our understanding of the relative pro-sociality of the rich. We apologize for this and have revised the paper accordingly.

To clarify, the aim of our study was never to explain the inconsistent findings in the literature. We believe that the vast methodological differences in how socio-economic status (SES) and pro-sociality are measured in previous studies, as well as in the samples employed (from psychology students to nationally representative samples), coupled with the relatively small number of existing studies prevent one from drawing safe conclusions about the

causes of the inconsistent findings.¹ We regret if the previous version was not sufficiently clear on this important point.

What *was* the aim of our study? And, most importantly, why do we believe our findings make an important contribution to the literature? To answer the latter question, it is helpful to provide a summary of the recent findings. As we write in the (substantially revised) second paragraph of our paper (we have underlined studies that we did not refer to in the original submission):

“After early findings from psychological research received considerable media attention for intimating that the rich are excessively selfish [Piff et al. 2010, Piff et al. 2012, Guinote et al. 2015], the balance of evidence appears to have shifted. Scholars have questioned the external validity of the early results [Francis 2012, Trautman et al. 2013, Korndorfer et al. 2015], and two attempts to replicate the findings in [Piff et al. 2012] failed [Balakrishnan et al. 2017, Clerke et al. 2018]. More recently, studies have either found no clear link between wealth and pro-sociality [Trautman et al. 2013, Côté et al. 2015, Dubois et al. 2015] or, in the majority of cases, that the rich behave more pro-socially than the less well-off [Korndorfer et al. 2015, Falk et al. 2018, Schmukle et al. 2019, Kosse et al. 2020, Falk et al. 2020, Smeets et al. 2015, Benenson et al. 2007, Bauer et al. 2014, Angerer et al. 2015]. The fact that a positive relation between wealth and pro-sociality has been consistently found in large, nationally representative samples from several countries across the world (30+ countries in [Korndorfer et al. 2015, Schmukle et al. 2019], and 76 countries in [Falk et al. 2018]) would seem to cast doubt on the accuracy of the “selfish rich” stereotype.”

While it is seemingly impossible to explain the reasons for any contradictory findings – perhaps the early results were due to the specific subject pools studied, the subjective measures of social class employed, or due to a Type-I error arising from the small samples – what seems clear is that the *balance of evidence* suggests that there is no empirical support that the rich are more selfish than the poor. While we do not wish to go into details about all these studies here, a few things may be worth pointing out to support our conclusion regarding the current balance of evidence. The first is the failed replication by Balakrishnan et al. (2017) of the findings in Piff et al. (2012) in four high-powered, pre-registered studies

¹ For these reasons, we prefer not to attempt an extensive discussion of the particulars of the designs in previous studies in our paper. If you and the editor think this is important, we can add a discussion in the Supplementary Notes.

(see also the replication by Clerke et al. 2018).² The second is the recently published study by Schmukle et al. (2019) who use some of the largest datasets to date on the topic (Study 1: $N=27,714$ US households; Study 3: $N=30,985$ participants with representative samples from 30 countries). Not only did Schmukle et al. fail to replicate the finding of Côte et al. (2015) about the moderating effect of income inequality on the relative pro-sociality of the poor (which you refer to in your comment below), but they consistently found a *positive* relationship between SES and pro-sociality. And this is not the first large-scale study to find such relationship. Korndörfer et al. (2015) used large and representative samples across several countries and domains of pro-sociality (for a total of 8 studies; in some of them $N>30,000$). Finally, the recently published study by Falk et al. (2018) with a sample of 80,000 people from 76 countries (nationally representative samples in most cases) also documents a clear *positive* relationship between wealth and pro-sociality (although that is not the focus of their analysis).

The aim of our paper was not to explain inconsistent findings per se, but to explore whether the findings seemingly exonerating the rich depend on the specific methodologies employed by using a novel method that overcomes many of the problems of incentivized experiments and survey studies. As we now write in the third paragraph of our paper:

“While, at first pass, the current balance of evidence would seem to contradict the stereotypical view of the rich, upon reflection, one may have reservations about the correct interpretation of the data. A concern with the studies showing that the rich behave less selfishly is their reliance on self-reported measures of pro-social behavior and the insufficient control of relative incentives in the experiments. While these measures have several advantages, they also have important limitations when it comes to exploring the relative pro-sociality of the rich. For instance, rich individuals may give away more money in incentivized experiments not because they are less selfish, but because they need the money less than poor individuals do. Or, they may be more prone to lie about the extent of their pro-social behavior in surveys, especially, since they are likely to be under greater pressure to appear unselfish than poorer individuals. In other words, the measures used in previous studies cannot rule out alternative explanations for the findings. A methodological approach that would overcome the aforementioned concerns, therefore, could be of great value for advancing our understanding on this important topic.”

² Balakrishnan et al. (2017) also report the findings from a small-scale meta-analysis using data from five studies employing the same design as Piff et al. (2012), including the original study. The authors find no evidence of an association between SES and pro-sociality. In fact, they estimate a cross-study effect of 0.01 with a 95% Confidence Interval of $[-0.07, 0.05]$. For comparison, Piff et al. (2012) found an effect of -0.24 with a 95% Confidence Interval of $[-0.44, -0.04]$.

Finally, why do we believe our findings are important? As we explain in the paper, our finding that the rich do not behave more selfishly than the poor in our experiment is in line with evidence from nationally representative samples in the Netherlands and in several other countries (see Korndörfer et al. 2015; Schmukle et al. 2019; Trautmann et al. 2013). In this sense, our findings help alleviate concerns about the interpretation of the data in the above-mentioned studies which relied on self-reported measures of pro-sociality obtained under scrutiny. To that end, in the concluding section of our paper, we summarize the importance of our findings as follows:

“Although no empirical study can provide by itself a definitive answer to a question as general as whether the rich are more selfish than the less well-off, by virtue of this being the first field experiment on the topic, we believe our findings make an important contribution to advancing our understanding. Specifically, our methodology, unlike that in previous studies, allows us to rule out alternative interpretations of the data. By showing that the rich do not behave more selfishly than the poor in a natural setting — thus avoiding “demand effects” and “social desirability bias” — while controlling the relative incentives for behaving pro-socially (see BTC treatment) and avoiding issues of self-selection, our study helps alleviate concerns about how the fast-growing body of evidence from surveys and incentivized experiments contradicting the stereotype of the “selfish rich” should be interpreted. In summary, while more field experiments are clearly needed, in light of our findings, the current balance of evidence, would seem to suggest the rich are not more selfish than the less well-off.”

We apologize for not being sufficiently clear in the original submission and thank you for your comment. As the passages above show, we have revised our paper considerably to ensure that its aim and contribution are clearly stated.

(continued from above)

As far as I see, settings are considerably different as are the point of departures and the variables included. For example, the finding by, e.g. Côté, House and Willer (2015, mentioned also by the authors) is on giving in the presence of income inequality. This is quite a different angle. In the present study, no control for inequality is included (e.g. inequality of the neighborhood would have been an option). So, the article fails to make clear why a field experiment can solve the issue - it actually does not. This is my main concern.

Thank you for this suggestion. As mentioned in our response to your previous comment, the evidence about the mediating role of income inequality on the relative pro-sociality of the rich is at best mixed. In a recent paper, Schmukle et al. (2019) present results from three large datasets in an attempt to replicate the findings in Côté et al. 2015. Schmukle et al. (2019) find no evidence of income equality playing a mediating role. In the authors' words: "To summarize, we did not find the postulated interaction between household income and state-level inequality on generosity in any of our analyses, although our sample sizes ($n = 27,714$ and $n = 43,739$) were 18- and 29-fold larger, respectively, than the sample size ($n = 1,498$) of Côté et al. (14)."

Nevertheless, in light of your comment, we collected new data from CBS Netherlands concerning income inequality at the neighborhood level. This is official data, meaning that we had zero degrees of freedom about how to assign households into neighborhoods or calculating income inequality. We find that controlling for income inequality in our regression analysis does *not* affect our results: the rich are substantially more likely to return envelopes even after controlling for income inequality in the neighborhood. We have added a discussion in the Results section, while the respective analysis is presented in Model (VI) in Table 1.

We hope the additional data and analysis has alleviated your concern.

Minor points:

- the setting seems to be quite artificial: who sends cash in a semi-transparent envelope? I am surprised that this worked so well.

Let us start by explaining why we chose to use semi-transparent envelopes. We did this such

that we could alter the incentives in a controlled manner. Audiences that have seen us present our experiment have always been very appreciative of this feature of our design. Reviewer 2 also remarked on this design choice in his/her report: "... the use of translucent envelopes with readable messages and visible cash or bank transfer card (BTC) contents is a clever way to vary the personal and social incentives for re-posting the letter."

You are raising an interesting point. Should one be concerned that the use of semi-transparent envelopes may affect our estimates? One should be concerned if two conditions are simultaneously satisfied: (i) the envelopes make individuals suspicious, and (ii) suspicion affects the rich and poor differently. With regards to (i), we note that neither semi-transparent envelopes nor misdelivered letters are uncommon in the Netherlands. The key question then for answering (i) is the following: Would *anyone* send money in a semi-transparent envelope? Our answer is as follows. Most people understand that some people do extraordinary things, i.e., things that a large majority of people would not do.³ As long as there is a reasonable possibility that *someone* could do X, there is no reason to be suspicious. This is why we decided to have the sender be a grandfather; a person that may recall more innocent times and be unaware of the dangers of sending money via post.

Anecdotal evidence from our experiment does not hint to participants becoming suspicious. Some individuals ($N=16$) returned our envelopes after placing them in non-transparent A4-sized envelopes. All of them were accompanied by letters explaining that the mailman made a mistake (and usually added that Joost – the proper recipient – should advise his grandfather not to send money via post). None of these letters suggested that those returning the envelopes were suspicious. Still, in light of the above, even if some people became suspicious, we would expect them to be a small fraction of our sample. Therefore, we have no reason to expect that they would have a noticeable effect on our estimates, even if we had reasons to expect the poor and the rich to be differentially affected – which we do not.

We added a discussion along these lines in the Supplementary Note 3. We hope we have addressed your concern.

³ For example, most people do not drive when heavily drunk, but clearly some do. Most people do not put their lives at risk to take a selfie, but again some clearly do.

Minor points:

- The discussion about the costs of posting the letter is overstretched: there are many mailboxes. The behavior under study is a relatively small prosocial act towards an anonymous stranger.

We agree that the cost of re-posting the letter is likely to be small in terms of time and effort; we never meant to suggest otherwise. In fact, we regard this as an advantage of our design as it makes for an easier comparison of the behavior of the rich and poor. We now clarify this in footnote 1 in the introduction in which we write:

“We purposefully chose a pro-social act (returning undelivered letters) that would not be too costly. This way, any differences between the rich and poor are likely to be minimal. In the Supplementary Note 5, we present evidence that survey respondents estimated they would need three minutes, on average, to repost a undelivered letter.”

Note that we did not make any statement about how high (or low) these costs may be in the original (or current) version of our manuscript. What we did was to point toward the evidence that, as the time from one’s last payday passes, the poor are less likely to return BTC envelopes (see discussion about Table 1). Since there is no change in the benefit from returning the envelope, we interpret this as an increase in costs owing to the increased financial pressure the poor are under. This interpretation would be in line with existing evidence in the literature about the impact of financial pressure on the behavior of the poor.⁴

Minor points:

- Why are the results for the 5 euro and 20 euro not presented separately? Also, effects of the control variables are just reported as insignificant but not shown. On the other hand, insignificant results (see the interaction effect in table 1) are reported and interpreted.

These are both good points.

⁴ The general intuition can perhaps be best conveyed through an example. The time and effort it takes to respond to an email from a family member or a friend is usually small – perhaps a few minutes. However, for an individual who is facing pressing deadlines at work, the “subjective” cost (what economists would call the *opportunity* cost) could be high.

With regards to the results from the 5 and 20 Euro conditions, we originally had this analysis in the paper. However, since the separate analyses and discussion took up considerable space without providing any substantive insights, in the end, we decided to pool the data from these conditions. To clarify, we obtain the same results if, instead of pooling the data, we perform our analysis separately for envelopes with 5 Euros and 20 Euros. We should have placed this analysis in the *Supplementary Information* but neglected to do so. The analysis can now be found in Supplementary Note 3.

With regards to the presentation of our estimates, you are right that there was an unintended asymmetry in how the estimates were presented. This arose from the fact that our controls were so numerous that they took up lots of space. In light of your comment, however, as well as some other comments we received, we decided to change the way we present our regression estimates. We now present and discuss all estimates for our main variables, i.e., variables for which we have clear hypotheses, in the main body of the paper. All empirical models that include additional controls (i.e., for variables obtained from CBS for which however we do not have clear ex ante hypothesis) are placed in Supplementary Table 3, where the reader can see the coefficients for all variables.

We hope we have addressed your comment in a satisfactory manner.

Minor points:

- The online supplementary material is already on the web and dates back to August 2017, and presents more extensive analyses. Also, I found the study (in a version of 2017) on econweb. Is the study already published? There is also a YouTube clip about the study, stressing actually that poor and rich differ only in their behavior but not in their preferences. See here.

There are two parts in this comment. The first one is straightforward. Our paper has never been published in a journal. An early draft was circulated as an NBER working paper. This is common practice in economics.⁵ We also note that having a working paper circulated online is perfectly in line with the submission guidelines of *Nature Communications*. This is especially true since, as you noted, the 2017 version is very different from the present version.

The second comment requires a bit more elaboration. Indeed, as we notified the editor at the time of our submission, our paper has received quite a bit of attention by the media. For

⁵ One can easily check this claim, e.g., by checking papers of NBER members such as John List (U of Chicago). In fact, a high fraction of NBER working papers – probably the most prestigious working paper series in economics – ends up being published in leading, international, peer-reviewed journals.

instance, apart from the YouTube clip you mention, Freakonomics Radio did a podcast that had been downloaded more than 1.5M times at the time of our first submission. Again, this is not against the policies of *Nature Communications*. It is true that, at one point in time, we believed the difference in behavior did not reflect a difference in underlying preferences between rich and poor. This conclusion was based on the estimates of a structural (econometric) model, which you can find in the aforementioned working paper. However, further analysis revealed that this conclusion depended on an assumption about the cost function (modelling the cost for reposting letters) that could not be empirically validated. Under alternative assumptions our structural estimates suggested a significant difference in pro-sociality between the rich and the poor. As a result, we dropped this part of our paper. After all, it was not crucial for answering our main question of whether there was support for “the selfish rich” stereotype. In the current version (as well as in the original submission), we abstain from making any claims about the underlying *preferences* of the rich and the poor. Instead, we focus on whether we find support for the stereotype. Once our paper is accepted for publication, we intend to share our manuscript with the outlets where our paper was featured. They will have the option to replace the content or add an addendum.

Minor points:

- The variables included in the regression model could be explained and justified more systematically. E.g., it is not argued in the theory section that the number of weeks since pay day is expected to matter.

This is a good point. In hindsight, we see that this part could have been written better. In light of your comment, we rewrote this part considerably. We now provide explicit justification and hypotheses for the variables included in our analysis in the paper.

Minor points:

- Is there also a letter sent without money, be it cash or bank transfer? Why are the results not reported?

There is no such treatment. All treatments that we conducted are reported in the main body of the paper. Similarly, all observations collected are used in the analysis.

Minor points:

- Are sure that foreigners could read the letter? In poor areas the number of foreigners is higher.

This is another good point. In fact, the concern extends beyond whether the non-Dutch individuals can read the letter. It is possible, for example, that the non-Dutch recipients of our letter feel less connection with (or empathy toward) the intended recipient (who as the name reveals is Dutch), than the Dutch recipients. Hence, they may be less likely to return the letter than Dutch recipients, thus exaggerating the difference between rich and poor.⁶

To explore this possibility, we replicated our empirical analysis after dropping the 27 households that had one or more non-Dutch members. This leaves us with 333 households. Our estimates are very similar to those obtained using the entire sample and our conclusions remain unchanged. The results from this analysis can be found in a new column (Model IV) that we added in Table 1.

Minor points:

- The ancillary study where respondents got monetary incentives for posting a letter does actually show no difference between rich and poor. It also does not study postal services. It does study what happens if you benefit yourself (in contrast to others).

We agree with both points. First, after reading your comment, we realized that the following sentence we wrote in the original submission was confusing: “Specifically, as part of a study on the use of the postal services, ...”. We have reworded the statement as follows: “We justified this request [to post an envelope in exchange for 20 Euros] by saying that we were studying the use of postal services.” We hope this clarifies.

Second, we do mention that in this treatment there are no differences in return rates between rich and poor: “Poor and rich households posted 60% and 69% of the envelopes, respectively (N=90, p=0.51, two-tailed, Fisher-exact).”

⁶ While we tried to avoid sampling non-Dutch households (see the Supplementary Note 1), it is true that some non-Dutch recipients ended up in our sample, and they are predominantly poor.

Minor points:

- Studies in the references 35) and 36) are wrongly cited, they are not about rich and poor neighborhoods. Also, these studies deliberately address passengers in a neighborhood and not households.

Thank you for pointing out this issue to us.

The main aim of study [35] is to look at return rates of dropped letters where the authors exogenously varied where the letter was placed (either on the street or behind a wind-shield wiper), and to whom the letter was addressed (a native Dutch person or a non-native Moroccan/Turkish person). The authors did include neighborhood wealth as a control variable, but wealth was not included as a design feature (it is in our study). As for study [36], we did not wish to imply that it has anything to do with neighborhood wealth or data at the household level. We just wanted to mention evidence consistent with the idea that the quality of a neighborhood's social environment can affect pro-social behavior, and that such quality is likely to be lower in poor neighborhoods. We have changed the passage to clarify these points. The passage is as follows:

“Another study hypothesizes and finds that letters “lost” (either dropped by the experimenters on the street or placed on the windshield of a car) in more affluent neighborhoods in the Netherlands were more likely to be returned [35]. Although the evidence in [35] is in line with our findings, it is difficult to make inferences about the relative prosociality of the rich and the poor for two reasons. First, neighborhood wealth is likely to be positively correlated with the quality of the social environment. In turn, the quality of the social environment is known to positively affect norm compliance [36]. Second, a well-known problem with the “lost letter technique” is that it is unclear who finds the letter. This, in fact, was the original inspiration for the “misdirected letter technique” [27] that we use in our experiment.”

Again, we thank you for your careful reading of our submission and helpful comments. We hope you will find the revised version suitable for publication at *Nature Communications*.

Response to the comments made by Reviewer 2

We are thankful for your careful reading of our manuscript and thoughtful comments. We have addressed them all. We hope you will be satisfied with the changes we made to the manuscript in response to your comments and that you will find the new version to be suitable for publication in *Nature Communications*.

Below, we present your comments in boxes, followed by our responses.

There is some controversy in the literature regarding the relation between socioeconomic status and prosocial behavior. There is evidence that low SES persons are more prosocial than their high SES counterparts; as the current authors point out, there is also contrary evidence. There have been attempts to resolve this discrepancy in findings, arguing that the SES-prosocial behavior relation is moderated by factors such as degree of economic inequality, or whether the prosocial behavior in question is public or private in nature.

Rather than addressing these moderating factors, the current authors appear to cast doubt on the validity of prior findings (they mention small, convenience samples; self-reported behavior, etc., although such issues do not apply to all of the relevant prior research) and propose that their study is free of such issues. The study they report is certainly interesting and methodologically ingenious. Whether it settles the question of how SES is related to prosocial behavior is open to question, however.

Thank you for your kind words. We were very pleased to learn that you found our study to be “interesting and methodologically ingenious”. Your comment indicates that our original submission was not sufficiently clear on a couple of important points such as the aim of our experiment and why we believe our findings are important. In hindsight, we can understand why this happened and take full responsibility. We regret this and appreciate the opportunity to revise our manuscript.

We never meant to suggest that our paper *settles* the question of how SES is related to prosocial behavior, although, looking back at the original submission, we can see how certain passages could be misinterpreted. To be clear, we do not believe that any single paper can singlehandedly and definitively answer such a complex empirical question as ours. Rather, we believe it is the *balance of accumulating evidence* that can answer a question. We have revised carefully the Introduction and Discussion sections to explain this. For instance, in the Discussion we now state:

“Although no empirical study can provide by itself a definitive answer to a question as general as whether the rich are more selfish than the less well-off, by virtue of this being the first field experiment on the topic, we believe our findings make an important contribution to advancing our understanding. Specifically, our methodology, unlike that in previous studies, allows us to rule out alternative interpretations of the data. By showing that the rich do not behave more selfishly than the poor in a natural setting — thus avoiding “demand effects” and “social desirability bias” — while controlling the relative incentives for behaving pro-socially (see BTC treatment) and avoiding issues of self-selection, our study helps alleviate concerns about how the fast-growing body of evidence from surveys and incentivized experiments contradicting the stereotype of the “selfish rich” should be interpreted. In summary, while more field experiments are clearly needed, in light of our findings, the current balance of evidence, would seem to suggest the rich are not more selfish than the less well-off.”

In light of your comment, we also rewrote the part of the introduction in which we discuss the existing evidence in the literature and the motivation for our study. Here is an excerpt (we have underlined studies that we did not refer to in the original submission):

“After early findings from psychological research received considerable media attention for intimating that the rich are excessively selfish [Piff et al. 2010, Piff et al. 2012, Guinote et al. 2015], the balance of evidence appears to have shifted. Scholars have questioned the external validity of the early results [Francis 2012, Trautman et al. 2013, Korndorfer et al. 2015], and two attempts to replicate the findings in [Piff et al. 2012] failed [Balakrishnan et al. 2017, Clerke et al. 2018]. More recently, studies have either found no clear link between wealth and pro-sociality [Trautman et al. 2013, Côté et al. 2015, Dubois et al. 2015] or, in the majority of cases, that the rich behave more pro-socially than the less well-off [Korndorfer et al. 2015, Falk et al. 2018, Schmukle et al. 2019, Kosse et al. 2020, Falk et al. 2020, Smeets et al. 2015, Benenson et al. 2007, Bauer et al. 2014, Angerer et al. 2015]...

While, at first pass, the current balance of evidence would seem to contradict the stereotypical view of the rich, upon reflection, one may have reservations about the correct interpretation of the data. A concern with the studies showing that the rich behave less selfishly is their reliance on self-reported measures of pro-social behavior and the insufficient control of relative incentives in the experiments. ... In other words, the measures used in

previous studies cannot rule out alternative explanations for the findings. A methodological approach that would overcome the aforementioned concerns, therefore, could be of great value for advancing our understanding on this important topic.”

We hope we have addressed your concern. Thank you for giving us the opportunity to explain better the contribution of our study.

Starting with the strengths of the research, the 'misdirected letter' methodology has the advantage of being unobtrusive, removing (or at least minimizing) the influence of social desirability factors. Furthermore, the use of translucent envelopes with readable messages and visible cash or bank transfer card (BTC) contents is a clever way to vary the personal and social incentives for re-posting the letter. The sampling of households by property value, verified by CBS household income data, is an effective way of ensuring that the households to which the misdirected letters were 'posted' differed markedly in wealth. The findings appear to be clear-cut: in all conditions, richer households were more likely than poorer households to re-post the misdirected letters, although there was some evidence that this difference was moderated by whether the envelope contained cash or a BTC.

Turning to the limitations of the study, my main concern is that this is a study conducted in one city in one country, using a very specific measure of prosocial behavior. The authors suggest (in the title of their paper) that their results reject the stereotype of the 'selfish rich'. In their Discussion, they go so far as to claim that "our findings indicate the stereotype of the selfish rich is inaccurate" – a very general claim that fails to acknowledge the specificities of their research.

Thank you for all the positive remarks on our design and findings (“a clever way to vary the personal and social incentives for re-posting the letter”; “an effective way of ensuring that the households to which the misdirected letters were 'posted' differed markedly in wealth”; “the findings appear to be clear-cut”). You are making a few important points in this passage that we need to address.

First, your main concern relates to the extent to which our findings can be generalized to different contexts (“one city in one country”) and different measures of prosocial behavior. This concern is well taken. Clearly, the previous version of our paper was not sufficiently clear on this important point. We are strong believers that any kind of empirical findings – whether they are obtained in the lab, the field, or from surveys – should be generalized with great caution. Ultimately, whether a result is likely to generalize is an empirical question that

is best answered through the balance of evidence. Ideally, in light of our findings, what the scientific community would need is: (i) more evidence of prosocial behavior in the Netherlands; and (ii) similar evidence from different countries. We have re-written the Discussion section to include the following passage, in response your comment:

“The methodology used in the present study, of course, is not without limitations. A field experiment such as ours is resource-intensive. As a result, it cannot easily be scaled to nationally representative samples or to consider different kinds of pro-social behaviors. This raises a question that is all-too-familiar to experimental social scientists: To what extent can we expect the findings from this specific experiment to generalize (i) to different social contexts, and (ii) to different populations? With regards to (i), the propensity to return envelopes in “misdirected letter experiments” has been linked with giving in dictator games [Franzen and Pointer, 2013]. The latter is arguably the most commonly used method in incentivized experiments for measuring altruism, i.e., the extent to which an individual cares about the welfare of others [Piff et al. 2010, Piff et al. 2012, Côté et al. 2015, Korndorfer et al. 2015, Falk et al. 2018, Schmukle et al. 2019, Kosse et al 2020, Falk et al. 2020, Smeets et al. 2015]. Altruism, in turn, has been linked to the propensity of individuals to volunteer, help strangers, and donate money [Falk et al. 2018]. With regards to (ii), researchers have previously used surveys and incentivized experiments to study the relative pro-sociality of the rich in the Netherlands, i.e., where our experiment also took place. They did this using nationally representative samples [Trautman et al. 2013, Korndorfer et al. 2015], and different measures of pro-sociality [Trautman et al. 2013, Korndorfer et al. 2015, Smeets et al. 2015]. Like with our experiment, none of these studies found evidence that the rich behave more selfishly than the poor. In fact, on balance, the evidence shows the rich behave less selfishly: they are not more likely to lie, steal, or accept bribes [Trautman et al. 2013], but they volunteer more, trust more, help more, give more to charity [Trautman et al. 2013, Korndorfer et al. 2015] than the less well-off. Importantly, similar findings are obtained in representative samples in other countries around the world [Korndorfer et al. 2015, Falk et al. 2018, Schmukle et al. 2019]. Taken together, therefore, these findings suggest that there are no obvious reasons to expect that our results will not generalize to other social contexts or populations. Of course, the question of generalizability is ultimately an empirical question that can only be answered with more field experiments.”

We hope this addresses your first comment in the passage above. Second, in your comment you refer to the original title of our paper “Rejecting the ‘selfish rich’ stereotype in a field

experiment”. We believe the title was factually accurate since we specified that this was done “in *a* field experiment”. However, in light of your comment and to avoid confusing readers, we decided to change the title to the following: “Are the rich more selfish than the poor? A field-experimental test of a common stereotype”. The idea is that we state the motivating research question (the stereotype in question format) without offering an answer. We hope you like this better.

To what extent does the misdirected letter technique measure prosocial behavior? The beneficiary is remote and the need for help is quite abstract. The measure may to some degree reflect differences in institutional trust (I am not convinced by the ancillary findings relating to this issue based on a sample of university students, who are unlikely to reflect the sociodemographic attributes of the study sample). More importantly, perhaps, given the claims in the paper, there is the issue of the extent to which 'selfishness' is captured by this measure. Is failing to re-post a letter a 'selfish' act, especially in the conditions where there was no possible benefit to self by not doing so (other than the saving in time and effort)?

Thank you for this comment. Again, you are making several points worth addressing.

First, as mentioned above, the propensity to return envelopes in “misdirected letter experiments” has been linked (at the individual level) with the propensity to give money in a laboratory dictator game (Franzen and Pointer 2013). That is, those who give more are more likely to return misdelivered letters. The dictator game is arguably the most commonly used method for measuring altruism and selfishness in lab experiments both in economics and psychology. Indeed, Piff et al. (2010, 2012) used this game too. We now mention this explicitly in the Discussion section (see quoted passage on previous page).

Second, with regards to our ancillary findings on institutional trust, upon re-reading the relevant passage in our original submission we realized that some important information was unfortunately omitted, which may explain your skepticism. We need to make a few clarifications here. The first is that, as one might expect, our university sample is wealthier than the Dutch population. We know this as we used the standard World-Value-Survey question on income (V238). This approach enables us to compare directly wealth in our sample with that in the representative Dutch sample. Crucially, we have observations for *all* income categories in our sample (probably due to the existence of scholarships for poorer families). This information permits us to use population weights in our analysis to adjust our estimates. We have re-written the relevant passage, which now reads as follows:

“We conducted a survey with Dutch-only students at Erasmus University (N=133). Our sample includes respondents from a broad range of socio-economic backgrounds, from the poorest to the richest families (see Supplementary Note 6). After adjusting our estimates using population weights, we find a positive correlation between wealth and trust in the government ($p < 0.10$, two-tailed, Ordered Probit), the police ($p < 0.05$, two-tailed, Ordered Probit), and the press ($p < 0.05$, two-tailed, Ordered Probit). However, we do not find a relationship between wealth and trust in the postal service ($p = 0.54$, two-tailed, Ordered Probit).”

Finally, you are asking whether it makes sense to talk of selfishness when the envelope contains a bank card instead of money. We think it does, because by not reposting the individual saves time and effort at the expense of the intended recipient of the letter.

We hope we have addressed all the points in your comment in a satisfactory manner.

As Table 1 in the Supplementary Information shows, the rich households differed from the poor ones in several ways other than income/wealth: On average, the rich households are older, have larger families, are less ethnically diverse, and receive more pensions. I appreciate that the authors controlled for these (and other) differences in their regression analyses, but I doubt whether such statistical controls can remove the effect of the rich households simply having more spare time available.

It is actually not clear that the rich have more spare time than others in the Netherlands. A recent study found that the wealthy report “spending the same amount of time on overall leisure as the general population (46.3% and 45.8%, respectively)” (Smeets et al. 2019). The same applies in our sample. While the rich in our sample receive more pensions (which could suggest they have more spare time), the poor are more likely to receive unemployment benefits. In fact, the share of households that receives either unemployment benefits or a pension is about the same for the rich (0.49) and the poor (0.45) (Fisher exact test, $N_1 = 180$, $N_2 = 180$, $p = 0.53$). We added a brief discussion on this point in Supplementary Note 2.

Then there is the issue of the specific location of the research: one medium-sized city in one European country, a country with relatively low economic inequality as measured by the Gini coefficient (28.6%, versus 47% in the USA, according to the World Bank). Given the evidence that the SES-prosocial behavior relation is moderated by inequality, this strikes me as a potentially serious limitation.

This is an important point. First, we note that the evidence about the mediating role of income inequality on the relative pro-sociality of the rich is at best mixed. In a recently published paper, Schmukle et al. (2019) present results from three large datasets and find no evidence of income equality playing a mediating role. In the authors' own words: "in large representative datasets from the United States (study 1), Germany (study 2), and 30 countries (study 3), we did not find any evidence [that higher income individuals are less generous than poorer individuals]. Instead, our results suggest that the rich are not less generous than the poor, even when economic inequality is large." The authors also note the following: "To summarize, we did not find the postulated interaction between household income and state-level inequality on generosity in any of our analyses, although our sample sizes ($n = 27,714$ and $n = 43,739$) were 18- and 29-fold larger, respectively, than the sample size ($n = 1,498$) of Côté et al. (14)."

Nevertheless, following a comment by Reviewer 1, we collected new data from CBS Netherlands concerning income inequality at the neighborhood level. This is official data, meaning that we had zero degrees of freedom about how to assign households into neighborhoods or calculating income inequality. We find that controlling for income inequality in our regression analysis does *not* affect our results: the rich are substantially more likely to return envelopes even after controlling for income inequality in the neighborhood. We have added a discussion in the Results section, while the new estimates are presented in Model (VI) in Table 1.

Of course, the more general question of generalizability still lingers. We hope you will find that the revised discussion section (partly quoted earlier in this letter) does a better job at explaining why our findings should be of interest to a broad audience despite being from "one medium-sized city in one European country".

In summary, what we have here is a well-conducted study showing that in this specific social context, rich households are more likely than poor ones to re-post a misdirected letter. But before anyone can reasonably claim to be able to reject the stereotype of the selfish rich, evidence from a wider variety of contexts using a broader range of measures is surely needed.

As our revised Discussion clarifies, we fully agree that “before anyone can reasonably claim to be able to reject the stereotype of the selfish rich, evidence from a wider variety of contexts using a broader range of measures is surely needed”. In light of the existing evidence, we think that studies relying on field designs such as ours where selection, social desirability and demand effect bias are absent are particularly useful. Nevertheless, we believe our findings are important and of broad interest as they help tip the current balance of evidence to suggest that the stereotype of the “selfish rich” is not accurate as early studies would have one believe. Thank you once again for the careful evaluation of our work. We believe your comments helped improve our paper considerably. We hope you will agree.

New references:

Smeets, P., Whillans, A., Bekkers, R., & Norton, M. I. (2019). Time Use and Happiness of Millionaires: Evidence From the Netherlands. *Social Psychological and Personality Science*, forthcoming

Response to the comments made by Reviewer 3

Dear Prof. Franzen,

we are thankful for your careful reading of our manuscript, your thoughtful comments, and positive evaluation of our manuscript. We hope you will be satisfied with the changes we made to the manuscript to address your comments and that you will find the new version to be suitable for publication in *Nature Communications*.

Below, we present your comments in boxes, followed by our responses.

This is an excellent paper. It deals with a relevant and interesting question, is well written, straightforward, easy to comprehend, and supplemented with a number of ancillary tests. Congratulations to the authors! I enjoyed reading it. I have only a few minor comments and suggestions.

Thank you so much for your kind words and positive evaluation of our study. We are grateful.

1) The results displayed in Table 1 are based on a linear probability model. Are the results comparable if the analyses are conducted via logit or probit models? Could you include a sentence in the caption of Table 1 indicating the similarity (or difference) of results?

The estimates are very similar. That is, the choice of model does not affect our conclusions. For your convenience, we include the Probit estimates below. As requested, we have added a sentence in the caption of Table 1.

	I	II	III	IV	V	VI
Rich	0.43*** (0.04)	0.41*** (0.07)	0.36*** (0.08)	0.33*** (0.08)	0.27* (0.15)	0.34*** (0.11)
Cash		-0.20** (0.08)	-0.20** (0.09)	-0.21** (0.09)	-0.24** (0.10)	-0.21** (0.09)
Rich Cash		0.08 (0.10)	0.08 (0.10)	0.09 (0.10)	0.01 (0.26)	0.10 (0.11)
Week			-0.08** (0.03)	-0.09*** (0.04)	-0.09** (0.04)	-0.08** (0.04)
Week Rich			0.09* (0.05)	0.10* (0.05)	0.08 (0.11)	0.08 (0.05)
Distance Joost			-0.02 (0.01)	-0.02* (0.01)	0.00 (0.07)	-0.03 (0.02)
Distance Mailbox			-0.19*** (0.07)	-0.20*** (0.07)	0.06 (0.21)	-0.35* (0.20)
Gini						0.11 (0.91)
N	360	360	360	333	143	318
Sample restrictions	None	None	None	Dutch HH	1 Person HH	None
Pseudo R2	0.15	0.17	0.19	0.19	0.10	0.19

Table Estimates from a Probit model (marginal effects). The dependent variable is a dummy taking the value of 1 when an envelope is returned and 0 otherwise. ``Rich" and ``Cash" are dummy variables indicating whether an observation is associated with a rich household or the Cash treatment, respectively. ``Week" measures the number of weeks since the last payday. "Distance Joost" ("Distance Mailbox") presents the distance from a subject's household to the house that Joost lives (the nearest mailbox). Gini measures the Gini coefficient of the neighborhood of the subject's household. Standard errors are shown in parentheses (clustered at the street level). ***, **, * indicate significance at the .01, .05, and .10 levels, respectively.

2) Poor households often have a higher proportion of individuals with a migration background. The norm that misdelivered letters should be returned might be less widespread in such a population. Hence, you might compare two groups that would not only differ in wealth but also in their normative background. Can this possibility be excluded? If not (or yes) you might want to mention or discuss it shortly in the discussion section.

This is a very good point. We replicated our empirical analysis after dropping the 27 households that had one or more non-Dutch members. This leaves us with 333 households. Our estimates remain virtually unchanged. The results from this analysis can be found in a new column (Model IV) that we added in Table 1.

3) Also for the discussion section: The rich are obviously the winners in a given society. Therefore, it does not come as a surprise that they support the system, have higher trust in societal institutions, and show more norm conforming behavior, including following the norm to return a misdirected letter. Hence, higher norm compliance is a further candidate to answer the “why question”.

Excellent point! Thank you for the suggestion. We have added the following passage in the concluding paragraph of our paper:

“On the one hand, future studies can help understand the limits of our findings: under what conditions may the rich behave more selfishly than others? Such studies will help uncover why richer individuals may sometimes behave less selfishly than others. For instance, there may be social returns to kindness in some societies, or there may be a causal link between wealth and norm compliance that could be mediated through more positive life experiences, better education, etc.”

Once again, thank you for the careful reading of our manuscript, the helpful comments, and the positive evaluation.

Reviewers' Comments:

Reviewer #1:

Remarks to the Author:

Report

'The selfish rich'- revision

Reaction to author's comments

Thank you for clarifying on a number of issues raised. I will react in the following and also try to formulate the concerns that are still present.

1) Point of departure of the paper/research problem and relevance:

The authors admit that the point of departure in the paper was not clearly stated and that they did not mean to say that a field experiment solves the inconsistencies in previous studies. Rather, they state that they wanted to explore whether the earlier findings that show that the rich are more prosocial in their behavior are depending on the method (see the statement on page 3 in the reaction letter followed by a quote from the manuscript). However, what is the problem with the studies that show the positive association between being rich/resourceful and prosociality? E.g. what is a field experiment adding to studies like the one by Falk et al (2018) who employed a sample of 76.000 respondents in 80 countries? These studies measured prosociality with self-reports and not with behavioral data and furthermore, also social class was not always measured objectively, but studies used the subjective estimation. You mention that you are able to discuss the shortcomings of earlier literature in more detail – in order to make a clear contribution I think this is valuable. You state furthermore that the existing findings do not rule out alternative explanations. More in detail, in the lab experiments, the rich might have behaved prosocial because they did not need the incentives (and not because they are not selfish). Therefore, your treatment is with and without possible incentive (money vs cheque). However, your experiment does not rule out alternative explanations as well. For example, the behavior of the rich in your sample might not be a result of being rich, but of more conformation to existing social norms. This is quite plausible, also because you do not know who actually posted the letter: it could have been anyone in the household, not in particular the one who earns the money (and is rich). Hence, your experiment confirms earlier findings, but does not rule out multiple explanations.

In addition, the link between self-reports and actual behavior is shown in earlier studies (e.g. in the lost letter experiments by Koopmans or Volker et al.).

In summary, I find your point of departure still not convincing. I miss a more theoretical argument about the behavior of people with more resources. It is quite interesting that the rich behave more prosocially than poorer people. The stereotype says that the rich are selfish – if they are not, they should be equally prosocial than the poor. But they behave more prosocially and there is no explanation for this. Your unanswered explanatory question is why the poor are more selfish than the rich and the rich more prosocial than the poor.

2) Income inequality as an additional condition for selfish/prosocial behavior

The authors included CBS data in income inequality (btw: you state that you had no choice but to use the CBS index in order to control for inequality effects in addition to absolute income --- this sounds awkward. There are always degrees of freedom on how to measure inequality, e.g. the standard-deviation of the average income is gives already information). Anyway, inequality seems not to affect the results. In this regard, relative income would be an interesting measure because it would include neighborhood effects. This comment is actually a consequence of my more theoretical question: how can we be sure that the behavior under study is a consequence of being rich?

3) The strangeness of the semi-transparent envelop

You say that people do strange things. Indeed, this is often true. Still, one would like to know what people actually thought. I think it is in a study by Volker (2017?) in sociological science, where the researchers too interviews with people who posted the letter and they showed that in richer neighborhoods people wanted to establish social order.

4) Another point is about ethics. You mention that your study was approved, but using information about household income and combining this information with actual behavior requires actual informed consent, or not? Did the people know that they participated? Where they told afterwards?

5) Minor point: the list of references shows many inconsistencies.

Reviewer #2:

Remarks to the Author:

The authors have been very responsive to the reviewers' concerns. Although no single study is ever going to convince all readers, I do not think that the authors could reasonably be expected to have done more than they have. The work they report has clearly been done with great care, and the additional work they have done in preparing this revision has made a good paper even better. The main limitation of the work is that it remains a single study conducted in a single city in a single country. However, this limitation is now (more) clearly acknowledged.

The revised manuscript reads very well. Indeed, I have only one (very minor and rather nerdy) suggestion for a wording change: On p. 8, last paragraph, 4 lines from bottom, 'less' should read 'fewer'.

Antony Manstead

Reviewer #3:

Remarks to the Author:

The authors addressed all of my questions and comments very well in the revised manuscript. I am in favor of publishing the manuscript in Nature Communications.

Response to the comments made by Reviewer 1

We are thankful for your careful reading of our manuscript and comments. We have addressed them all. We hope that you will find the new version to be suitable for publication in *Nature Communications*.

Below, we present your comments in boxes, followed by our responses.

1) Point of departure of the paper/research problem and relevance:

The authors admit that the point of departure in the paper was not clearly stated and that they did not mean to say that a field experiment solves the inconsistencies in previous studies. Rather, they state that they wanted to explore whether the earlier findings that show that the rich are more prosocial in their behavior are depending on the method (see the statement on page 3 in the reaction letter followed by a quote from the manuscript). However, what is the problem with the studies that show the positive association between being rich/resourceful and prosociality? E.g. what is a field experiment adding to studies like the one by Falk et al (2018) who employed a sample of 76.000 respondents in 80 countries? These studies measured prosociality with self-reports and not with behavioral data and furthermore, also social class was not always measured objectively, but studies used the subjective estimation. You mention that you are able to discuss the shortcomings of earlier literature in more detail – in order to make a clear contribution I think this is valuable. You state furthermore that the existing findings do not rule out alternative explanations. More in detail, in the lab experiments, the rich might have behaved prosocial because they did not need the incentives (and not because they are not selfish). Therefore, your treatment is with and without possible incentive (money vs cheque). However, your experiment does not rule out alternative explanations as well. For example, the behavior of the rich in your sample might not be a result of being rich, but of more conformation to existing social norms. This is quite plausible, also because you do not know who actually posted the letter: it could have been anyone in the household, not in particular the one who earns the money (and is rich). Hence, your experiment confirms earlier findings, but does not rule out multiple explanations. In addition, the link between self-reports and actual behavior is shown in earlier studies (e.g. in the lost letter experiments by Koopmans or Volker et al.).

In summary, I find your point of departure still not convincing. I miss a more theoretical argument about the behavior of people with more resources. It is quite interesting that the rich behave more prosocially than poorer people. The stereotype says that the rich are selfish – if they are not, they should be equally prosocial than the poor. But they behave more prosocially and there is no explanation for this. Your unanswered explanatory question is why the poor are more selfish than the rich and the rich more prosocial than the poor.

Thank you for your comment. Given its length and the various issues you touch upon, we will divide our response into three parts, each time repeating the relevant part of your comment in *italics*, followed by our response to that passage in normal fonts. We will start with the final paragraph which summarizes your comment and work backwards.

In summary, I find your point of departure still not convincing. I miss a more theoretical argument about the behavior of people with more resources. It is quite interesting that the rich behave more prosocially than poorer people. The stereotype says that the rich are selfish – if they are not, they should be equally prosocial than the poor. But they behave more prosocially and there is no explanation for this. Your unanswered explanatory question is why the poor are more selfish than the rich and the rich more prosocial than the poor.

We agree with you about the importance of theoretical arguments. Indeed, already in the previous version, we provided two reasons why the rich behave more pro-socially in our experiment. Your comment suggests we failed to make these discussions memorable to the reader. In light of this, we added a new paragraph at the end of the Introduction where we discuss these issues (the theoretical framework and the two factors) prominently. We also added a brief passage in the first paragraph of the Discussion.

We would like to elaborate more on our point of departure, why we believe our paper is of relevance, and how we adjusted the manuscript. Our point of departure is the existence of a widely-held stereotype and the lack of clear evidence on its accuracy. In particular, the “selfish rich” stereotype that has been linked to preferences for “taxing the wealthy” and heightened social tensions. In other words, it is a stereotype with significant socio-economic consequences. Therefore, it is important to know if it is accurate. The existing evidence does not permit us to judge the accuracy of the stereotype for the reasons mentioned in the introduction. Our study’s main contribution is to test the accuracy of this stereotype using a novel method that overcomes many of the concerns found in previous studies. Specifically, we use a method that is unobtrusive and does not rely on subjective measures of social class. We do not find support for the stereotype; if anything, the rich behave more pro-socially. As we discuss in the paper, this finding is in line with the balance of evidence from survey studies.

The question that follows naturally is *why* the rich may behave more prosocially in our experiment. Given your comment, we ought to stress that *none* of the previous survey studies cited in our paper and finding wealth to be positively associated with pro-sociality attempted to provide an answer to this question to the best of our knowledge. We believe that the reason for this is that it is exceedingly difficult to offer a comprehensive account for the relationship between wealth and pro-sociality. Nevertheless, our data suggest two reasons for the differences. The first is that the rich value the money in the envelopes less than the poor, a phenomenon economists refer to as the “diminishing marginal utility of income.” We discuss

this in several places in the paper: in the first paragraph in the Results section, in the first full paragraph on page 7, in the opening paragraph in the Discussion (new addition), and also in the new paragraph closing out the Introduction. The second reason is that the poor experience more financial pressure than the rich, which reduces the propensity to carry out pro-social tasks. This explanation is consistent with the findings by Mani et al. (2012). We discuss this in the second paragraph of the Results section, in the opening paragraph in the Discussion (new addition), as well as in the new paragraph at the end of the Introduction.

To illustrate how important a theoretical perspective is, we spent a considerable amount of time building and estimating a formal mathematical model to test the significance of the aforementioned two factors in our data. We ultimately abandoned that approach in order to appeal to a more general audience. Nonetheless, all the insights we obtained from the formal model are included in our paper in a more generally accessible style befitting the academically diverse audience of *Nature Communications*. The details of the model can be found in the NBER Working Paper the purpose of which – like with all working paper series – was to encourage academic discussion prior to publishing our results.

We do not claim that these reasons are the only ones for the observed differences; they are not. Indeed, we state this clearly in the last paragraph in the Introduction, in the last paragraph in our paper, and in the middle of page 7.¹ We do not even wish to claim that the reasons discussed in the last paragraph of our paper are all the possible reasons. As we note at the end of the Introduction. Wealth is such a central force in modern societies that it arguably affects all aspects of our lives (e.g., education, life experiences, attitudes, norm compliance and more) making it exceedingly difficult to identify the mechanisms behind any behavioral differences.

[Y]our experiment does not rule out alternative explanations as well. For example, the behavior of the rich in your sample might not be a result of being rich, but of more conformation to existing social norms. This is quite plausible, also because you do not know who actually posted the letter: it could have been anyone in the household, not in particular the one who earns the money (and is rich). Hence, your experiment confirms earlier findings, but does not rule out multiple explanations. In addition, the link between self-reports and actual behavior is shown in earlier studies (e.g. in the lost letter experiments by Koopmans or Volker et al.).

As we state in the paper, the behavior of the rich could be due to a number of reasons including a greater adherence to norms (see last paragraph in the Discussion section). This, however, does not contradict the claim that they do not *behave* more selfishly: if they care more about *social* norms, they cannot behave more selfishly. Second, in the paper, we acknowledge the

¹ Specifically, on page 7, we added the following sentence: “A number of factors could account for this sizable difference such as differences in wealth, education, and social status between rich and poor. While our data do not permit us to identify the mechanism behind the difference, the evidence is clearly at odds with the stereotypical view of the rich.”

complexity that arises from having many people living inside a house. It is for this reason that, in the previous version of our paper, we showed that we obtain very similar results when we restrict our data analysis to one-person households (see Model V, Table 1). To highlight this point, we added a sentence on page 7 where we discuss Model V. (“This also ensures that the person returning the envelope is the income earner and not someone else living in the household.”) In any case, the fact that we do take these points into consideration illustrates that we never claimed to have identified all the mechanisms behind our results. This was not the aim of our study.

As for your reference to the studies by Koopmans and Veit (2014) and Volker et al. (2016), we would like to note the following. First, we added a reference to the former study. We didn’t cite this study previously because it relies on a German sample and does not focus on the relationship between wealth and pro-sociality.² Second, as we note in the paper, while these are related studies, given that it is unclear who finds the lost letters, one cannot make inferences about the relative pro-sociality of the rich – indeed, these papers do not attempt to do so. Hence, we respectfully disagree that they show the link between studies relying on self-reports and actual behavior.

The authors admit that the point of departure in the paper was not clearly stated and that they did not mean to say that a field experiment solves the inconsistencies in previous studies. Rather, they state that they wanted to explore whether the earlier findings that show that the rich are more prosocial in their behavior are depending on the method (see the statement on page 3 in the reaction letter followed by a quote from the manuscript). However, what is the problem with the studies that show the positive association between being rich/resourceful and prosociality? E.g. what is a field experiment adding to studies like the one by Falk et al (2018) who employed a sample of 76.000 respondents in 80 countries? These studies measured prosociality with self-reports and not with behavioral data and furthermore, also social class was not always measured objectively, but studies used the subjective estimation. You mention that you are able to discuss the shortcomings of earlier literature in more detail – in order to make a clear contribution I think this is valuable.

In this passage, you are describing two of the attractive, novel features of our study. As you note, there are several substantive differences in the methodology employed in our study and that used in studies such as that by Falk et al. (2018) and the preceding empirical studies on the topic of our investigation. The preceding empirical studies differ in so many dimensions from each other that a detailed discussion would require a separate paper. Instead, we prefer to focus on discussing some of the features they share and distinguish them from ours. The

² The authors control for the extent of unemployment in a neighborhood in their analysis but this is a noisy measure of wealth which moreover is not at the level of the household.

main difference, in our opinion, is that, unlike in previous studies, our experiment does not rely on self-reports of pro-sociality or monetary payments which fail to control for relative incentives. This is the main difference in the sense that, *all previous empirical studies* share these features. Moreover, these features are particularly problematic as they could systematically bias inferences. As we write in the third paragraph of our paper: “*A concern with the studies showing that the rich behave less selfishly is their reliance on self-reported measures of pro-social behavior and the insufficient control of relative incentives in the experiments... For instance, rich individuals may give away more money in incentivized experiments not because they are less selfish, but because they need the money less than poor individuals do. Or, they may be more prone to lie about the extent of their pro-social behavior in surveys, especially, since they are likely to be under greater pressure to appear unselfish than poorer individuals. In other words, the measures used in previous studies cannot rule out alternative explanations for the findings.*”

You mention another unique feature of our study, which is that our analysis makes use of an objective measure of wealth. Indeed, to the best of our knowledge, ours is the only study to use wealth/income data obtained by the tax authorities (who then share it with CBS Netherlands). This is what we currently state in the paper at the top of page 4: “[*W*]e are able to obtain household-level data from CBS Netherlands for our sample on a number of key socio-economic variables (e.g., wealth, ethnicity, household size). This allows us to check how wealthy a household is, perform randomization checks, explore the underlying differences between rich and poor, and investigate the robustness of our findings by adding controls in our regressions.” While we are fairly confident that all other empirical studies cited in our paper (e.g., Falk et al. 2018, Schmuckle et al. 2015, Trautmann, van de Kuilen, and Zeckhauser, 2013) rely on self-reported incomes, since this is not explicitly stated in their papers, we did not highlight this feature in our paper.^{3,4} However, if the editor would like us to do so, we would happily oblige.

We hope our response and the new paragraph in the Introduction (the last paragraph in that section) have addressed your comment.

2) Income inequality as an additional condition for selfish/prosocial behavior

The authors included CBS data in income inequality (btw: you state that you had no choice but to use the CBS index in order to control for inequality effects in addition to absolute income --- this sounds awkward. There are always degrees of freedom on how to measure inequality, e.g. the standard-deviation of the average income is gives already information).

³ We carefully read all papers (including supplementary information files) and, when needed, searched for information online to determine whether wealth was self-reported in each study. For instance, in the case of Trautmann et al. (2013) we had to check online to establish whether the panel they used to collect data (LISS Panel) obtained the data through CBS Netherlands or by using surveys (it was the latter).

⁴ In several instances, early studies often relied on convenience samples with little variation on real wealth (e.g., students), using priming tasks to induce the subjective perception of class. See Trautmann, van de Kuilen, and Zeckhauser (2013) for a discussion.

Anyway, inequality seems not to affect the results. In this regard, relative income would be an interesting measure because it would include neighborhood effects. This comment is actually a consequence of my more theoretical question: how can we be sure that the behavior under study is a consequence of being rich?

We are pleased to learn that our analysis seems to have convinced you that income inequality does not affect our results. Let us start by addressing the main point of your comment: controlling for relative income and neighborhood effects.

The editor asked us to perform the additional analysis on relative income you suggested. We acquired new data from CBS Netherlands on households' relative income. We added relative income as a control in our regression analysis. As can be seen in Table 1 (Model VII), this control has no impact on our main results. Interestingly, relative income has a *positive* and significant coefficient (at the 5-percent level) suggesting that the richer a household is in relative terms the more likely the envelope is to be returned. There could be different interpretations for this finding. One of them is that higher relative income does not create feelings of entitlement as argued by Coté et al. (2015), but feelings of social responsibility. We should also stress that to ensure our findings are robust to neighborhood effects, already from the first submission, we undertook to cluster standard errors at the street level. This was noted in Table 1, but to highlight it, in this version, we added a sentence at the end of the second paragraph of the Results section.

We note that the relative income in the new analysis is calculated at the district level. There were two reasons for this. The first and most important reason was that our data suggest the results are not sensitive to using neighborhood- or district-level data. Specifically, to address your comment on income inequality in the last round of revisions, we acquired data from CBS including information on both neighborhood and district level. We find that we obtain *identical* estimates and p-values for all our main variables if we use neighborhood or district-level controls. Given this, it made no sense to us to spend over twice the cost for neighborhood level data to obtain an identical result.

Finally, we would like to address two other passages in your comment. First, in your last sentence, you ask: "How can we be sure that the behavior under study is a consequence of being rich?" The answer is we cannot. We never claimed that the behavior we observe was a *consequence* of being rich. We cannot make that statement because people are not randomly assigned into wealth. The behavior *could be* a consequence of wealth, but it could also be the result of any of the other variables that are correlated with wealth (e.g., education, social status, positive life experiences). We believe this is clearly stated in our Discussion. (This is also clearly stated now in the new paragraph that concludes the Introduction.) For instance, as we write in the concluding paragraph of our paper:

“On the one hand, future studies can help understand the limits of our findings: under what conditions may the rich behave more selfishly than others? Such studies will help uncover why richer individuals may sometimes behave less selfishly than others. For instance, there may be social returns to kindness in some societies, or there may be a causal link between wealth and norm compliance that could be mediated through more positive life experiences, better education, etc.”

Second, we would like to address your parenthetical sentence. In the last round, you wrote the following:

“[T]he finding by, e.g. Coté, House and Willer (2015, mentioned also by the authors) is on giving in the presence of income inequality. This is quite a different angle. In the present study, no control for inequality is included (e.g. inequality of the neighborhood would have been an option)”

In light of your comment, we followed Coté et al. (2015) in using the Gini coefficient to control for income inequality. Here is what we wrote in our reply (emphasis added):

“[I]n light of your comment, we collected new data from CBS Netherlands concerning income inequality at the neighborhood level. This is official data, meaning that we had zero degrees of freedom about how to assign households into neighborhoods or calculating income inequality.”

We had zero degrees of freedom in the sense that CBS Netherlands did all this for us. We never said that we had zero degrees of freedom when it came to *measuring* inequality.

3) The strangeness of the semi-transparent envelop

You say that people do strange things. Indeed, this is often true. Still, one would like to know what people actually thought. I think it is in a study by Volker (2017?) in sociological science, where the researchers too interviews with people who posted the letter and they showed that in richer neighborhoods people wanted to establish social order.

We are pleased that our response on the issue of semi-transparent envelopes seems to have convinced you. We agree that it would be nice to know what people were thinking when receiving/returning the misdelivered envelopes, however, we believe that such surveys as the ones you propose have limitations as they suffer from social desirability bias. We are of the opinion that the best way to uncover motives for returning letters is to conduct incentivized

experiments. Indeed, as mentioned in our paper, there is such evidence showing a positive correlation between returning misdelivered envelopes and giving in dictator games -- a measure of altruism (Franzen and Pointer, 2013). Finally, we note that we read Volker (2017, *Sociological Science*)⁵ and we could not find a discussion of interviews. We also could not find any other paper by Volker that discussed such evidence.

4) Another point is about ethics. You mention that your study was approved, but using information about household income and combining this information with actual behavior requires actual informed consent, or not? Did the people know that they participated? Where they told afterwards?

Under the Federal Policy for the Protection of Human Subjects, the IRB may waive informed consent if the research poses no more than minimal risk, could not be carried out practicably without the waiver, and the waiver will not adversely affect the rights and welfare of the subjects. Even when information on household income and the like is included, a waiver can be granted (as was the case for our study) if the dataset does not include any identifiable information. The IRB that reviewed our protocol judged that all these conditions were met. CBS Netherlands also ruled that there was no way for us to identify individuals through the dataset.

As for debriefing, again, there are several instances when the request for debriefing can be waived. In our case, Dutch Mail took the position that debriefing could have reputational costs that exceed the benefits, and hence ruled in favor of not debriefing participants. We are not unique in this. In fact, we know that this is the very common for natural field experiments, at least in economics. But we wonder: did Volker et al. (2016), or Koopmans and Veit (2014) debrief subjects who found lost letters? Did Kaizer et al. (2007, *Science*) or Volker (2017) debrief subjects? One would argue this would be especially important in the “disorder treatments” where the researchers littered public spaces giving participants the impression of a disorderly neighborhood? Balafoutas, Nikiforakis and Rockenbach (2016, *Nature Communications*) certainly did not debrief participants, in line with IRB protocols.

5) Minor point: the list of references shows many inconsistencies.

⁵ Volker, B. (2017). Revisiting broken windows: The role of neighborhood and individual characteristics in reaction to disorder cues. *Sociological science*, 4, 528-551.

Thank you. We have fixed them.

Response to the comments made by Reviewer 2

Dear Prof. Manstead,

We are thankful for your careful reading of our manuscript and comments which helped improve our work. We are grateful for your positive assessment of our revision and recommending the publication of our paper in *Nature Communications*.

Reviewer #2 (Remarks to the Author):

The authors have been very responsive to the reviewers' concerns. Although no single study is ever going to convince all readers, I do not think that the authors could reasonably be expected to have done more than they have. The work they report has clearly been done with great care, and the additional work they have done in preparing this revision has made a good paper even better. The main limitation of the work is that it remains a single study conducted in a single city in a single country. However, this limitation is now (more) clearly acknowledged. The revised manuscript reads very well.

We are most grateful for your kind words and positive evaluation of our paper.

Indeed, I have only one (very minor and rather nerdy) suggestion for a wording change: On p. 8, last paragraph, 4 lines from bottom, 'less' should read 'fewer'.

Antony Manstead

We have made the change. Thank you.

Response to the comments made by Reviewer 3

Dear Prof. Franzen,

We are thankful for your careful reading of our manuscript and comments which helped improve our work. We are grateful for your positive assessment of our revision and recommending the publication of our paper in *Nature Communications*.

Reviewer #3 (Remarks to the Author):

The authors addressed all of my questions and comments very well in the revised manuscript. I am in favor of publishing the manuscript in *Nature Communications*.

We are most grateful for your positive recommendation.